# Learning Disentangled Representation for Multi-Modal Time-Series Data

## Abstract

Multi-modal time series data is common in web technologies like the Internet of Things (IoT). Existing methods for multi-modal time series representation learning aim to disentangle the modality-shared and modality-specific latent variables. Although achieving notable performances on downstream tasks, they usually assume an orthogonal latent space. However, the modality-specific and modality-shared latent variables might be dependent on real-world scenarios. Therefore, we propose a general generation process, where the modality-shared and modality-specific latent variables are dependent, and further develop a **M**ulti-mod**A**l **TE**mporal Disentanglement (**MATE**) model. Specifically, our **MATE** model is built on a temporally variational inference architecture with the modality-shared and modality-specific prior networks for the disentanglement of latent variables. Furthermore, we establish identifiability results to show that the extracted representation is disentangled. More specifically, we first achieve the subspace identifiability for modality-shared and modality-specific latent variables by leveraging the pairing of multi-modal data. Then we establish the component-wise identifiability of modality-specific latent variables by employing sufficient changes of historical latent variables. Extensive experimental studies on 12 datasets show a general improvement in different downstream tasks, highlighting the effectiveness of our method in real-world scenarios.

## CCS Concepts

• **Do Not Use This Code** → **Generate the Correct Terms for Your Paper**; *Generate the Correct Terms for Your Paper*; Generate the Correct Terms for Your Paper; Generate the Correct Terms for Your Paper.

## Keywords

Multimodal Time Series, Time Series Representation

**ACM Reference Format:**
. 2018. Learning Disentangled Representation for Multi-Modal Time-Series Data. In *Proceedings of Make sure to enter the correct conference title from your rights confirmation emai (Conference acronym 'XX)*. ACM, New York, NY, USA, 18 pages. https://doi.org/XXXXXXX.XXXXXXX

## 1 Introduction

The World Wide Web plays a critical role in generating and processing time series data, from web traffic [60, 69] to IoT device outputs

**Unpublished working draft. Not for distribution.**

[46, 67]. By providing access to real-time data through APIs [2] and analytics tools, it enables comprehensive analysis and visualization of trends. Time series forecasting allows the web to power intelligent services such as microservice log analysis [5] and IoT systems [8], where sensors continuously generate data streams that predict future behaviors, monitor device performance, and trigger early warning alerts for emerging issues.

Most of the existing works for time series analysis [52, 58, 68, 95, 109, 114] are devised for homogeneous data, with the assumption that time series are sampled from the same modality. However, the heterogeneous time series data [62, 64, 85], which are sampled from multiple modalities and not compatible with these methods, are also common in several real-world applications, e.g., Internet of Things (IoT) [73, 78, 97], health care [33, 70, 107], and finance [6, 113]. To model the multi-modal time series data, one mainstream solution is to disentangle the modality-specific and modality-shared latent variables from the observational time series signal.

Several methods are proposed to disentangle the modality-specific and modality-shared temporally latent variables. One mainstream approach is based on the contrastive learning method. For example, Deldari et.al proposes COCOA [12], which learns modality-shared representations by aligning the representation from the same timestamp, and Ouyang et.al propose Cosmo [75], which extracts modality-shared representations by using a iterative fusion learning strategy. Considering that the modality-specific representations also play an important role in the downstream task, Liu et.al [64] use an orthogonality restriction and simultaneously leverage the modality-shared and modality-specific representations. Considering the multi-view setting as a special case of the multi-modal setting, Huang et.al [26] develop the identifiability results of the latent temporal process by minimizing the contrastive objective function. In summary, these methods usually assume that the modality-shared and modality-specific latent variables are orthogonal, hence they can be disentangled by using different contrastive-learning-based constraints.

Although these methods achieve outstanding performance on several applications, the orthogonality of modality-shared and modality-specific latent space may be too difficult to satisfy in real-world scenarios. Figure 1 provides an example of physiological indicators of diabetics, where brain-related and heart-related signals are observed in time series data. Specifically, Figure 1 (a) denotes the true data generation process, where the causal directions from insulin concentration to blood pressure and heart rate denote how diabetes leads to complications of heart disease and high blood pressure. As shown in Figure 1 (b), existing methods that apply orthogonal constraints on the estimated latent variables despite the dependent true latent sources, lead to the entanglement and further the suboptimal performance of downstream tasks.

To address the aforementioned challenge of dependent latent sources, we propose a multi-modal temporal disentanglement framework to estimate the ground-truth latent variables with identifiability guarantees. Specifically, we first leverage the pair-wise

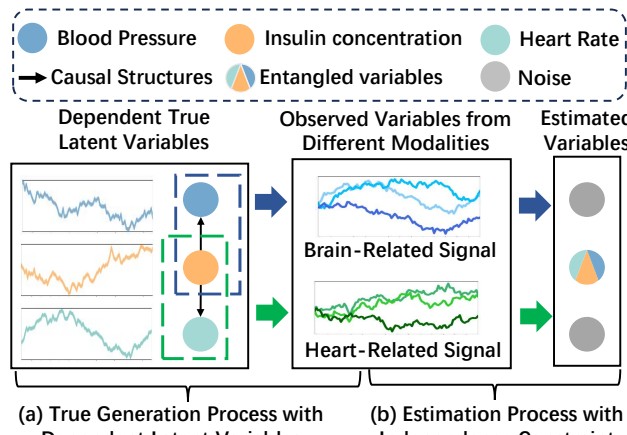

**Figure 1: Illustration of physiological indicators of diabetics, where brain-related and heart-related signals are observations. (a) In the true generation process, observations are generated from dependent latent sources. (b) In the estimation process, enforcing orthogonality on estimated sources can result in the entanglement of latent sources and meaningless noises.**

multi-modal data to establish the subspace identifiability of latent variables. Sequentially, we leverage the independent influence of historical latent variables to further show the component-wise identifiability of latent variables. Building on the theoretical results, we develop the **M**ulti-mod**A**l **TE**mporal Disentanglement (**MATE**) model, which incorporates variational inference neural architecture with modality-shared and modality-specific prior networks. The proposed **MATE** is validated through extensive downstream tasks for multi-modal time series data. The impressive performance that outperforms state-of-the-art methods demonstrates its effectiveness in real-world applications.

## 2 Related Works

### 2.1 Multi-modal Representation Learning

Multi-modal representation learning [37, 57, 63, 77, 92] aims to mean information from different modalities, and has lots of applications like Visual Question Answering (VQA) [16, 47, 53, 86, 99]. The mainstream methods include self-supervised learning [35, 65, 106], masked autoencoders [19, 22, 47], and the generative model-based methods [39, 59]. Multi-modal time series data is underexplored in literature, despite being often encountered in practice. One of the mainstream methods for multi-modal time series representation learning is to extract the modality-shared representation. Previously, Deldari et.al [12] extracted the modality-shared representation by computing the cross-correlation of different modalities and minimizing the similarity between irrelevant instances. Deng [13] proposes multi-modality data augmentation to learn inter-modality and intra-modality representations. Recently, Kara [38] devised a factorized multi-modal fusion mechanism for leveraging cross-modal correlations to learn modality-specific representations. And Liu et.al [64] leverage both the modality-shared and modality-specific representation for downstream tasks. However, most of

this method implicitly assumes that the latent space is orthogonal, which may be hard to meet in real-world scenarios. In this paper, we propose a data generation process with dependent subspace for mutli-modal time series data and devise a flexible model with theoretical guarantees.

### 2.2 Identifiability of Generative Model

To achieve identifiability [71, 80, 93] for causal representation, several researchers use the independent component analysis (ICA) to recover the latent variables with identification guarantees [20, 66, 84, 100]. Conventional methods assume a linear mixing function from the latent variables to the observed variables [7, 27, 50, 108]. Since the linear mixing process is hard to meet in real-world scenarios, recently, some researchers have established the identifiability via nonlinear ICA by using different types of assumptions like auxiliary variables or sparse generation process [28, 31, 41, 56, 112]. Specifically, Aapo et.al [29, 30, 32, 40] achieve the identifiability by assuming the latent sources with exponential family and introducing auxiliary variables e.g., domain indexes, time indexes, and class labels. And Zhang et.al [43, 45, 96, 98] achieve the component-wise identification results for nonlinear ICA without using the exponential family assumption. To achieve identifiability without any supervised signals, several researchers employ sparsity assumptions [28, 31, 41, 56, 112]. For example, Lachapelle et al. [48, 49] introduced mechanism sparsity regularization as an inductive bias to identify causal latent factors. And Zhang et.al [110] use the sparse structures of latent variables to achieve identifiability under distribution shift. Researchers also employ nonlinear ICA to achieve identifiability of time series data [21, 26, 61, 98]. For example, Aapo et.al [29]) adopt the independent sources premise and capitalize on the variability in variance across different data segments to achieve identifiability on nonstationary time series data. And Permutation-based contrastive learning is employed to identify the latent variables on stationary time series data. Recently, LEAP [103] and TDRL [102] have adopted the properties of independent noises and variability historical information. Song et.al [88] identify latent variables without observed domain variables. As for the identifiability of modality, Imant et.al [10] present the identifiability results for multimodal contrastive learning. Yao et.al [100] consider the identifiability of multi-view causal representation under the partially observed settings. In this paper, we leverage the pairwise of multi-modality data and variability historical information to achieve identifiability for multi-modality time series data.

## 3 Problem Setup

### 3.1 Data Generation Process of Multi-modal Time Series

To show how to learn disentangled representation for multi-modal time series data, we first introduce the data generation process as shown in Figure 2. Specifically, we assume that the existence of $M$ modalities $\mathcal{S} = \{S_1, S_2, \cdots, S_M\}$. For each modality $S_m$, time series data with discrete time steps $x^{s_m}_{1:T} = \{x^{s_m}_1, x^{s_m}_2, \cdots, x^{s_m}_T\}$ with the length of $T$ are drawn from a distinct distribution, represented as $p(x^{s_m}_{1:T})$. Moreover, $x^{s_m}_t$ is generated from the modality-shared and modality-specific latent variables $z^c_t, z^{s_m}_t$ by an invertible and

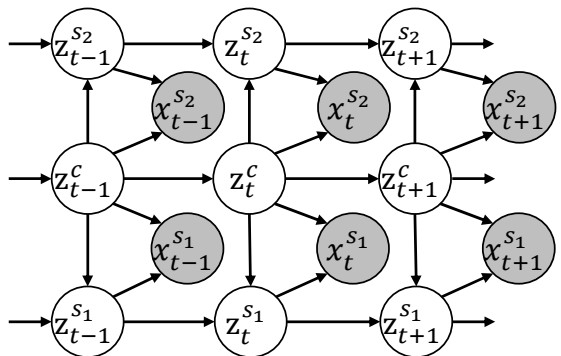

**Figure 2: Data generation process of time series data with two modalities. The grey and white nodes denote the observed and latent variables, respectively.**

nonlinear mixing function $g_m$ shown as follows:

$$x_t^{s_m} = g_m(z_t^c, z_t^{s_m}).\tag{1}$$

For convenience, we let $z_t^m = \{z_t^c, z_t^{s_m}\}$ be the latent variables of $m$-th modality. And we further let $z_t^c = (z_{t,i}^c)_{i=1}^{n_c}$ and $z_t^{s_m} = (z_{t,i}^{s_m})_{i=n_c+1}^n$. More specifically, the $i$-th dimension modality-shared latent variables $z_{t,i}$ are time-delayed and related to the historical modality-shared latent variables $z_{t-\tau}$ with the time lag of $\tau$ via a nonparametric function $f_i^c$. Similarly, the modality-specific latent variables are generated via another nonparametric function $f_i^m$, which are formalized as follows:

$$\begin{aligned} z_{t,i}^c &= f_i^c(PA(z_{t,i}^c), \epsilon_{t,i}^c), \quad \epsilon_{t,i}^c \sim p_{\epsilon_{t,i}^c} \\ z_{t,i}^{s_m} &= f_i^m(PA(z_{t,i}^{s_m}), \epsilon_{t,i}^{s_m}), \quad \epsilon_{t,i}^{s_m} \sim p_{\epsilon_{t,i}^{s_m}}, \end{aligned}\tag{2}$$

where $PA$ denote the set of latent variables that directly cause $z_{t,i}^c$ or $z_{t,i}^{s_m}$, and $\epsilon_{t,i}^{s_m}, \epsilon_{t,i}^c$ denote the independent noise. Combining the example of diabetics in Figure 1, $x_t^{s_1}$ and $x_t^{s_2}$ can be considered as brain-related and heart-related signals, respectively. The modality-shared variables $z_t^c$ denote the insulin concentration and $z_t^{s_1}, z_t^{s_2}$ denote the blood pressure and heart rate, respectively. $z_t^c \rightarrow \{z_t^{s_1}, z_t^{s_2}\}$ denotes that insulin concentration influences blood pressure and heart rate.

## 3.2 Problem Definition

Based on the aforementioned data generation process, we further provide the problem definition. Specifically, We are first supposed to have a set of $M$ sensory modalities. Then, for each group of time series from $M$ modalities, we let $y$ be the corresponding label. Given the labeled multi-modal time series training set with the size of $D$, i.e., $\{X_i, y_i\}_{i=1}^D$, we aim to obtain a model that can extract disentangled representations for multi-modal time series data, which can benefit the downstream tasks, i.e. estimate correct label. More mathematically, our goal is to estimate the distribution of the modality-specific latent variables $p(z_{1:T}^{s_1}), \cdots, p(z_{1:T}^{s_M})$ and the modality-shared latent variables $p(z_{1:T}^c)$ by modeling the observed

multi-modal time series data, which are formalized as follows:

$$\begin{aligned} &\ln p(x_{1:T}^{s_1}, \cdots, x_{1:T}^{s_M}) \\ &= \int_{z_{1:T}^{s_1}} \cdots \int_{z_{1:T}^{s_M}} \int_{z_{1:T}^c} \Big( \ln p(x_{1:T}^{s_1}, \cdots, x_{1:T}^{s_M} | z_{1:T}^{s_1}, \cdots, z_{1:T}^{s_M}, z_{1:T}^c) \\ &+ \sum_{m=1}^M \ln p(z_{1:T}^{s_m} | z_{1:T}^c) + \ln p(z_{1:T}^c) \Big) dz_{1:T}^{s_1} \cdots dz_{1:T}^{s_M} dz_{1:T}^c. \end{aligned}\tag{3}$$

Therefore, to achieve this goal, we first devise a temporal variational inference architecture with prior networks to reconstruct the modality-specific and modality-shared latent variables, which are shown in Section 4. Sequentially, we further propose theoretical analysis to show that these estimated modality-shared and modality-specific latent variables are identifiable, which are shown in Section 5.

## 4 MATE: Multi-modal Temporal Disentanglement Model

Based on the data generation process in Figure 2, we proposed the Multi-modal temporal Disentanglement (MATE) model as shown in Figure 3, which is built upon the variation auto-encoder. Moreover, it includes the shared prior networks and the private prior networks, which are used to preserve the dependence between the modality-specific and modality-shared latent variables. Furthermore, we devise a modality-shared constraint to enforce the invariance of modality-shared latent variables from different modalities.

### 4.1 Variational-Inference-based Neural Architecture

We begin with the evidence lower bound (ELBO) based on the proposed data generation process. Without loss of generality, we consider two modalities, i.e., $M = 2$, so the ELBO can be formalized as Equation (4). Please refer to Appendix B for the derivation details.

$$\begin{aligned} p(x_{1:T}^{s_1}, x_{1:T}^{s_2}) \geq \mathcal{L}_r &- \underbrace{D_{KL}(q(z_{1:T}^c | x_{1:T}^{s_1}, x_{1:T}^{s_2}) || p(z_{1:T}^c))}_{\mathcal{L}_c} \\ &- \underbrace{D_{KL}(q(z_{1:T}^{s_1} | x_{1:T}^{s_1}, z_{1:T}^c) || p(z_{1:T}^{s_1} | z_{1:T}^c))}_{\mathcal{L}_{s_1}} \\ &- \underbrace{D_{KL}(q(z_{1:T}^{s_2} | x_{1:T}^{s_2}, z_{1:T}^c) || p(z_{1:T}^{s_2} | z_{1:T}^c))}_{\mathcal{L}_{s_2}}, \end{aligned}\tag{4}$$

and $\mathcal{L}_r$ denotes the reconstruct loss and it can be formalized as:

$$\begin{aligned} \mathcal{L}_r =& \mathbb{E}_{q(z_{1:T}^{s_1} | x_{1:T}^{s_1}, z_{1:T}^c)} \mathbb{E}_{q(z_{1:T}^c | x_{1:T}^{s_1}, x_{1:T}^{s_2})} \ln p(x_{1:T}^{s_1} | z_{1:T}^{s_1}, z_{1:T}^c) \\ &+ \mathbb{E}_{q(z_{1:T}^{s_2} | x_{1:T}^{s_2}, z_{1:T}^c)} \mathbb{E}_{q(z_{1:T}^c | x_{1:T}^{s_1}, x_{1:T}^{s_2})} \ln p(x_{1:T}^{s_2} | z_{1:T}^{s_2}, z_{1:T}^c)), \end{aligned}\tag{5}$$

where $q(z_{1:T}^{s_1} | x_{1:T}^{s_1}, z_{1:T}^c), q(z_{1:T}^{s_2} | x_{1:T}^{s_2} z_{1:T}^c)$, and $q(z_{1:T}^c | x_{1:T}^{s_1}, x_{1:T}^{s_2})$ are used to approximate the prior distributions of modality-specific and modality-shared latent variables and are implemented by neural architecture based on convolution neural networks (CNNs). In practice, we devise a modality-specific encoder for each modality, which can be formalized as follows:

$$z_{1:T}^{s_1}, z_{1:T}^{c_1} = \psi_{s_1}(x_{1:T}^{s_1}), \quad z_{1:T}^{s_2}, z_{1:T}^{c_2} = \psi_{s_2}(x_{1:T}^{s_2}),\tag{6}$$

**Figure 3: Illustration of the proposed MATE model, we consider two modalities for a convenient understanding, more modalities can be easily extended. Modality-specific encoders are used to extract the latent variables of different modalities. The specific prior networks and the shared prior network are used to estimate the prior distribution for KL divergence.**

Moreover, since $z_{1:T}^{c_1}$ and $z_{1:T}^{c_2}$ should be as similar as possible, we further devise a modality-shared constraint as shown in Equation (7), which restricts the similarity of modality-shared latent variables between any two pairs of modalities.

$$\mathcal{L}_s = \sum_{s_i, s_j, \in \mathcal{S}, i \neq j} \log \frac{z_{1:T}^{c_{s_i}} \cdot z_{1:T}^{c_{s_j}}}{|z_{1:T}^{c_{s_i}}||z_{1:T}^{c_{s_j}}|} \tag{7}$$

By using the modality-shared constraint, we can simply let $z_{1:T}^c = z_{1:T}^{c_1}$ be the estimated modality-shared latent variables.

As for $p(x_{1:T}^{s_1}|z_{1:T}^{s_1}, z_{1:T}^c))$ and $p(x_{1:T}^{s_2}|z_{1:T}^{s_2}, z_{1:T}^c))$, which model the generation process from latent variables to observations via Multi-layer Perceptron networks (MLPs) as shown in Equation (8).

$$\hat{x}_{1:T}^{s_1} = \phi_{s_1}(z_{1:T}^{s_1}, z_{1:T}^c), \quad \hat{x}_{1:T}^{s_2} = \phi_{s_2}(z_{1:T}^{s_2}, z_{1:T}^c) \tag{8}$$

Finally, the $p(z_{1:T}^{s_1}|z_{1:T}^c), p(z_{1:T}^{s_2}|z_{1:T}^c)$ and $p(z_{1:T}^c)$ in Equation (4) denotes the prior distribution of latent variables, which are introduced in subsection 4.2. Please refer to Appendix D for more details on the architecture of the proposed **MATE** model.

## 4.2 Specific and Shared Prior Networks

**Shared Prior Networks for Modality-shared Estimation:** To model the shared prior distribution $p(z_{1:T}^c)$, we first review the transition function of shared latent variables in Equation (2). Without loss of generality, we consider the time-lag as 1, hence we let $\{r_i^c\}$ be a set of inverse transition functions that take $z_{t,i}^c, z_{t-1}^c$ as input and output the independent noise, i.e., $\epsilon_{t,i}^c = r_i^c(z_{t,i}^c, z_{t-1}^c)$. Note that these inverse transition functions can be implemented by simple MLPs. Sequentially, we devise a transformation $\sigma^c := \{\hat{z}_{t-1}^c, \hat{z}_t^c\} \rightarrow \{\hat{z}_{t-1}^c, \hat{\epsilon}_t^c\}$ and its corresponding Jacobian can be formalized as $\mathbf{J}_{\sigma^c} = \begin{pmatrix} \mathbb{I} & 0 \\ * & \text{diag}\left(\frac{\partial r_i^c}{\partial \hat{z}_{t,i}^c}\right) \end{pmatrix}$, where $*$ denotes a matrix. By applying the change of variables formula, we have the following

equation, we estimated the prior distribution as follows:

$$\log p(\hat{z}_{t-1}^c, \hat{z}_t^c) = \log p(\hat{z}_{t-1}^c, \hat{\epsilon}_t^c) + \log |\det(\mathbf{J}_{\sigma^c})|. \tag{9}$$

Moreover, we can rewrite Equation (9) to Equation (10) by using independent noise assumption.

$$\log p(\hat{z}_t^c|\hat{z}_{t-1}^c) = \log p(\hat{\epsilon}_t^c) + \sum_{i=1}^{n_c} \log |\frac{\partial r_i^c}{\partial \hat{z}_{t,i}^c}|. \tag{10}$$

As a result, the prior distribution shared latent variables can be estimated as follows:

$$p(\hat{z}_{1:T}^c) = p(\hat{z}_1^c) \prod_{\tau=2}^{T} \left( \sum_{i=1}^{n_c} \log p(\hat{\epsilon}_{\tau,i}^c) + \sum_{i=1}^{n_c} \log |\frac{\partial r_i^c}{\partial \hat{z}_{\tau,i}^c}| \right), \tag{11}$$

where $p(\hat{\epsilon}_{\tau,i}^c)$ is assumed to follow a standard Gaussian distribution.
**Private Prior Networks for Modality-private Prior Estimation:** We assign each modality an individual prior network and take modality $s_1$ as an example. Similar to the derivation of the shared prior networks, we let $\{r_i^{s_1}\}$ be a set of inverse transition functions that take $z_{t,i}^{s_1}, z_{t-1}^{s_1}$ and $z_t^c$ as input and output the independent noise, i.e., $\epsilon_{t,i}^{s_1} = r_i^{s_1}(z_{t,i}^{s_1}, z_{t-1}^{s_1}, z_t^c)$. Therefore, we can estimate the prior distribution of specific latent variables in a similar manner as shown in Equation (12).

$$p(\hat{z}_{1:T}^{s_1}|\hat{z}_{1:T}^c)$$
$$= p(\hat{z}_1^{s_1}|\hat{z}_{1:T}^c) \prod_{\tau=2}^{T} \left( \sum_{i=n_c+1}^{n} \log p(\hat{\epsilon}_{\tau,i}^{s_1}|\hat{z}_{1:T}^c) + \sum_{i=n_c+1}^{n} \log |\frac{\partial r_i^{s_1}}{\partial z_{\tau,i}^{s_1}}| \right). \tag{12}$$

## 4.3 Model Summary

By using the estimating private and shared priors to calculate the KL divergence in Equation (4), we can reconstruct the latent variables by modeling the observations from different modalities. Note that our method can be considered a flexible backbone architecture for multi-modal time series data, the learned latent variables can be applied to any downstream tasks. Therefore, by letting $\mathcal{L}_y$ be the objective function of a downstream task and combining Equation

(4) with the modality-shared constrain in Equation (7), the total loss of the proposed MATE model can be formalized as follows:

$$\mathcal{L}_{total} = -\alpha \mathcal{L}_r + \beta(\mathcal{L}_c + \mathcal{L}_{s_1} + \mathcal{L}_{s_2}) + \gamma \mathcal{L}_s + \mathcal{L}_y, \quad (13)$$

where $\alpha$, $\beta$ and $\gamma$ are hyper-parameters.

## 5 Theoretical Analysis

To show the proposed method can learn the disentangled representation, we first provide the definition of subspace and component-wise identifiability. We further provide theoretical analysis regarding identifiability. Specifically, we leverage nonlinear ICA to show the subspace-identifiability (Theorem 1) and component-wise identifiability (Corollary 1.1) of the proposed method.

### 5.1 Subspace Identifiability and Component-wise Identifiability

Before introducing the theoretical results about identifiability, we first provide a brief introduction to subspace identification and component-wise identification. As for subspace identification [55], the subspace identification of latent variables $z_t$ means that for each ground-truth $z_{t,i}$, there exits $\hat{z}_t$ and an invertible function $h_i : \mathbb{R}^n \to \mathbb{R}$, such that $z_{t,i} = h_i(\hat{z}_t)$. As for component-wise identifiability [45], the component-wise identifiability of $z_{t,i}$ means that for each ground-truth $z_{t,i}$, there exits $\hat{z}_{t,j}$ and an invertible function $h_i : \mathbb{R} \to \mathbb{R}$, such that $z_{t,i} = h_i(\hat{z}_{t,j})$. Note that the subspace identifiability provides a coarse-grained theoretical guarantee for representation learning, ensuring that all the information is preserved. While the component-wise identifiability provides a coarse fine theoretical guarantee, ensuring that the estimated and ground-truth latent variables are one-to-one corresponding.

### 5.2 Subspace Identifiability of Latent Variables

Based on the definition of latent causal process, we first show that the modality-shared and modality-specific latent variables are subspace identifiable, i.e., the estimated modality-shared latent variables $\hat{z}_t^c$ (modality-specific latent variables $\hat{z}_t^{s_m}$) contains all and only information in the true modality-shared latent variables $z_t^c$ (modality-specific latent variables $z_t^{s_m}$). Since the multi-modal time series data are pair-wise, without loss of generality, we consider modality $s_m$ as the example.

THEOREM 1. *(Subspace Identification of the Modality-shared and Modality-specific Latent Variables) Suppose that the observed data from different modalities is generated following the data generation process in Figure 2, and we further make the following assumptions:*

- *A1 (Smooth and Positive Density:) The probability density of latent variables is smooth and positive, i.e., $p(z_t|z_{t-1}) > 0$ over $\mathcal{Z}_t$ and $\mathcal{Z}_{t-1}$.*
- *A2 (Conditional Independence:) Conditioned on $z_{t-1}$, each $z_{t,i}^c$ is independent of $z_{t,j}^c$ for $i, j \in \{1, \cdots, n_c\}, i \neq j$. And conditioned on $z_{t-1}$ and $z_t^c$, each $z_{t,i}^{s_m}$ is independent of $z_{t,j}^{s_m}$, for $i, j \in \{n_c + 1, \cdots, n\}, i \neq j$.*
- *A3 (non-singular Jacobian): Each $g_m$ has non-singular Jacobian matrices almost anywhere and $g_m$ is invertible.*

- *A4 (Linear Independence:) For any $z_t^{s_*} \in \mathcal{Z}_t^{s_*}$, there exist $n_c + 1$ values of $z_{t-1,k}^{s_m}, k = n_c + 1, \cdots, n$, such that these vectors $v_{t,j}$ are linearly independent, where $v_{t,j,k}$ are defined as follows:*

$$v_{t,j} = \left( \frac{\partial^2 \log p(z_{t,j}^{s_m}|z_{t-1}^m, z_t^c)}{\partial z_{t,j}^{s_m} \partial z_{t-1,n_c+1}^{s_m}}, \cdots, \frac{\partial^2 \log p(z_{t,j}^{s_m}|z_{t-1}^m, z_t^c)}{\partial z_{t,j}^{s_m} \partial z_{t-1,n}^{s_m}} \right) \quad (14)$$

*Then if $\hat{g}_1 : \mathcal{Z}_t^c \times \mathcal{Z}_t^{s_1} \to \mathcal{X}_t^{s_1}$ and $\hat{g}_2 : \mathcal{Z}_t^c \times \mathcal{Z}_t^{s_2} \to \mathcal{X}_t^{s_2}$ assume the generating process of the true model $(g_1, g_2)$ and match the joint distribution $p(x_t^{s_1}, x_t^{s_2})$ of each time step then $z_t^c$ and $z_t^{s_m}$ are subspace identifiable.*

**Proof Sketch:** The proof can be found in Appendix A.1. First, we construct an invertible transformation $h_m$ between the ground-truth latent variables and estimated ones. Sequentially, we prove that the ground truth modality-shared latent variables are not the function of modality-specific latent variables by leveraging the pairing time series from different modalities. Sequentially, we leverage sufficient variability of historical information to show that the modality-specific latent variables are not the function of the estimated modality-shared latent variables. Moreover, by leveraging the invertibility of transformation $h_m$, we can obtain the Jacobian of $h_m$ as shown in Equation (15),

$$\mathbf{J}_{h_m} = \begin{bmatrix} \mathbf{A} := \frac{\partial z_t^c}{\partial \hat{z}_t^c} & \mathbf{B} := \frac{\partial z_t^c}{\partial \hat{z}_t^{s_m}} = 0 \\ \mathbf{C} := \frac{\partial z_t^{s_m}}{\partial \hat{z}_t^c} = 0 & \mathbf{D} := \frac{\partial z_t^{s_m}}{\partial \hat{z}_t^{s_m}}, \end{bmatrix} \quad (15)$$

where $B = 0$ and $C = 0$, since the ground truth modality-shared latent variables are not the function of modality-specific latent variables and the modality-specific latent variables are not the function of the estimated modality-shared latent variables, respectively.

**Discussion of the Assumptions:** The proof can be found in Appendix A.1. The first and the second assumptions are common in the existing identification results [102, 103]. The third assumption is also common in [44], meaning that the influence from each latent source to observation is independence. The final assumption means that the historical information changes sufficiently, which can be easily satisfied with sufficient time series data.

### 5.3 Component-wise Identifiability of Latent Variables

Based on Theorem 1, we further establish the component-wise identifiability result as follows.

COROLLARY 1.1. *(Component-wise Identification of the Modality-shared and Modality-specific Latent Variables) Suppose that the observed data from different modalities is generated following the data generation process in Figure 2, and we further make the assumptions A1, A2 and the following assumptions:*

- *A5 (Linear Independence:) For any $z_t \in \mathcal{Z}_t$, there exist $2n+1$ values of $z_{t-1,k}^m, k = 1, \cdots, n$, such that these vectors $v_{t,l}$ are linearly*

**Table 1: Time series classification for Motion, D1NAMO, WIFI, and KETI datasets.**

| Model | Motion | | DINAMO | | WIFI | | KETI | |
|---|---|---|---|---|---|---|---|---|
| | Accuracy | Macro-F1 | Accuracy | Macro-F1 | Accuracy | Macro-F1 | Accuracy | Macro-F1 |
| ResNet | 89.96(0.234) | 91.41(0.139) | 88.64(0.262) | 88.58(0.273) | 90.29(0.519) | 88.14(0.648) | 96.05(0.387) | 84.59(1.181) |
| MaCNN | 85.57(2.117) | 86.93(2.429) | 90.17(0.172) | 48.56(1.666) | 88.81(3.821) | 87.80(3.353) | 93.05(1.411) | 71.93(2.178) |
| SenenHAR | 88.95(0.369) | 88.66(0.276) | 89.56(0.620) | 47.23(0.182) | 94.63(0.614) | 92.75(0.686) | 96.43(0.143) | 84.74(0.379) |
| STFNets | 89.07(0.098) | 88.84(0.229) | 90.51(0.450) | 47.50(0.132) | 80.52(0.245) | 75.93(1.262) | 89.21(0.808) | 69.55(0.476) |
| RFNet-base | 89.93(0.281) | 91.70(0.408) | 90.76(0.252) | 58.79(4.911) | 86.31(1.765) | 82.56(2.313) | 95.12(0.478) | 81.45(1.077) |
| THAT | 89.66(0.488) | 91.38(0.521) | 92.76(0.292) | 71.64(2.229) | 95.59(1.027) | 94.86(1.126) | 96.33(0.283) | 85.12(1.143) |
| LaxCat | 60.25(3.678) | 41.01(4.381) | 90.64(0.362) | 54.56(2.013) | 76.36(1.492) | 73.85(2.155) | 93.33(1.449) | 70.67(0.335) |
| UniTS | 91.02(0.399) | 92.73(0.432) | 90.88(0.362) | 58.39(4.048) | 95.83(0.812) | 94.49(1.383) | 96.04(0.613) | 84.08(1.601) |
| COCOA | 88.31(0.254) | 89.27(0.702) | 90.69(0.189) | 55.00(1.495) | 87.76(0.531) | 84.51(0.728) | 92.68(1.062) | 74.72(1.987) |
| FOCAL | 89.37(0.083) | 90.91(0.191) | 90.52(0.220) | 52.00(2.104) | 94.15(0.208) | 92.68(0.377) | 94.88(0.371) | 78.47(1.043) |
| CroSSL | 91.32(0.992) | 89.94(1.353) | 91.05(0.438) | 53.13(0.781) | 76.80(2.206) | 68.45(3.054) | 93.63(0.504) | 76.25(1.538) |
| MATE | **92.44(0.160)** | **93.75(0.154)** | **93.31(0.170)** | **73.72(1.148)** | **96.95(0.231)** | **96.20(0.431)** | **97.00(0.097)** | **86.93(0.924)** |

**Table 2: Time series classification for human motion prediction and healthcare datasets.**

| Model | HumanEVA | | H36M | | UCIHAR | | MIT-BIH | |
|---|---|---|---|---|---|---|---|---|
| | Accuracy | Macro-F1 | Accuracy | Macro-F1 | Accuracy | Macro-F1 | Accuracy | Macro-F1 |
| ResNet | 86.68(0.327) | 86.51(0.247) | 92.44(0.278) | 92.27(0.289) | 93.12(0.630) | 93.01(0.637) | 98.52(0.066) | 97.62(0.083) |
| MaCNN | 86.27(0.047) | 86.12(0.041) | 78.54(0.430) | 77.73(0.647) | 84.57(0.851) | 84.06(0.936) | 97.26(0.186) | 96.07(0.194) |
| SenenHAR | 85.77(1.078) | 86.00(1.185) | 67.69(0.525) | 67.44(0.490) | 87.77(1.228) | 87.47(1.252) | 95.82(0.036) | 94.79(0.735) |
| STFNets | 86.00(0.368) | 85.76(0.291) | 61.67(1.481) | 57.20(1.112) | 81.64(0.521) | 81.64(0.339) | 91.63(0.369) | 88.97(0.217) |
| RFNet-base | 97.15(0.616) | 96.18(0.457) | 94.14(0.674) | 93.14(0.710) | 95.63(0.952) | 95.16(1.414) | 98.64(0.139) | 97.85(0.108) |
| THAT | 85.95(0.226) | 85.90(0.207) | 81.28(0.351) | 81.27(0.182) | 93.06(0.364) | 93.06(0.422) | 98.49(0.159) | 97.56(0.237) |
| LaxCat | 86.28(0.023) | 86.20(0.041) | 86.09(2.516) | 85.84(2.495) | 89.00(0.476) | 88.78(0.429) | 97.77(0.113) | 96.77(0.131) |
| UniTS | 97.90(0.561) | 97.52(0.879) | 94.96(0.461) | 94.81(0.152) | 94.75(0.526) | 94.72(0.528) | 98.75(0.078) | 97.95(0.099) |
| COCOA | 93.46(0.293) | 91.63(1.469) | 84.12(1.670) | 83.85(1.820) | 94.11(0.425) | 93.96(0.616) | 97.76(0.241) | 96.64(0.979) |
| FOCAL | 92.15(1.428) | 91.83(1.214) | 89.73(0.270) | 89.30(0.282) | 94.36(0.098) | 94.36(0.190) | 98.67(0.053) | 97.84(0.103) |
| CroSSL | 86.29(0.045) | 86.06(0.273) | 87.35(1.447) | 83.62(1.546) | 94.45(0.170) | 93.83(0.530) | 97.96(0.167) | 95.06(0.071) |
| MATE | **98.90(0.108)** | **98.82(0.094)** | **96.12(0.036)** | **95.99(0.037)** | **95.97(0.258)** | **95.93(0.273)** | **98.97(0.065)** | **98.34(0.147)** |

independent, where $v_{t,l}$ are defined as follows:

$$v_{t,l} = \left( \frac{\partial^3 \log p(z_{t,l}^c|z_{t-1}^m)}{\partial^2 z_{t,l}^c \partial z_{t-1,1}^m}, \cdots, \frac{\partial^3 \log p(z_{t,l}^c|z_{t-1}^m)}{\partial^2 z_{t,l}^c \partial z_{t-1,n}^m}, \right.$$

$$\frac{\partial^2 \log p(z_{t,l}^c|z_{t-1}^m)}{\partial z_{t,l}^c \partial z_{t-1,1}^m}, \cdots, \frac{\partial^2 \log p(z_{t,l}^c|z_{t-1}^m)}{\partial z_{t,l}^c \partial z_{t-1,n}^m},$$

$$\frac{\partial^3 \log p(z_{t,l}^{sm}|z_{t-1}^m, z_t^c)}{\partial^2 z_{t,l}^{sm} \partial z_{t-1,1}^m}, \cdots, \frac{\partial^3 \log p(z_{t,l}^{sm}|z_{t-1}^m, z_t^c)}{\partial^2 z_{t,l}^{sm} \partial z_{t-1,n}^m}, \tag{16}$$

$$\left. \frac{\partial^2 \log p(z_{t,l}^{sm}|z_{t-1}^m, z_t^c)}{\partial z_{t,l}^{sm} \partial z_{t-1,1}^m}, \cdots, \frac{\partial^2 \log p(z_{t,l}^{sm}|z_{t-1}^m, z_t^c)}{\partial z_{t,l}^{sm} \partial z_{t-1,n}^m} \right)$$

Then if $\hat{g}_1 : \mathcal{Z}_t^c \times \mathcal{Z}_t^{s_1} \to \mathcal{X}_t^{s_1}$ and $\hat{g}_2 : \mathcal{Z}_t^c \times \mathcal{Z}_t^{s_2} \to \mathcal{X}_t^{s_2}$ assume the generating process of the true model $(g_1, g_2)$ and match the joint distribution $p(x_t^{s_1}, x_t^{s_2})$ of each time step then $z_t^c$ is component-wise identifiable.

**Proof Sketch and Discussion:** The proof can be found in Appendix A.2. Based on Theorem 1, we employ similar assumptions like [102, 103] to construct a full-rank linear system with only zero solution, which ensures the component-wise identifiability of latent variables, i.e., the estimated and ground truth latent variables are one-to-one corresponding.

## 5.4 Relationships between Identifiability and Representation Learning

Intuitively, the proposed method is more general since existing methods with orthogonal latent space are a special case of the data generation process shown in Figure 2. We further discuss how these identifiability results benefit the representation learning for multi-modal time-series sensing signals. First, the subspace identifiability results show that the modality-shared and modality-specific latent

variables are disentangled under the dependent latent process, naturally boosting the downstream tasks that require modality-shared representations. Second, the component-wise identifiability result uncovers the latent causal mechanisms of multi-modal time series data, which potentially provides the interpretability for multi-modal representation learning, i.e., finding the unobserved confounders. Third, by identifying the latent variables, we can further model the data generation process, which enhances the robustness of the representation of multi-modal time series sensing signals.

**Table 3: Time series classification for audio and video dataset.**

| Model | HAC | | | | EPIC-Kitchens | |
|---|---|---|---|---|---|---|
| | Human | | Cartoon | | D2 | |
| | Accuracy | Macro-F1 | Accuracy | Macro-F1 | Accuracy | Macro-F1 |
| ResNet | 93.93(0.462) | 93.90(0.475) | 88.22(0.574) | 88.04(0.938) | 76.75(0.066) | 76.35(0.456) |
| RFNet-base | 93.47(0.886) | 93.51(0.862) | 86.54(0.941) | 86.09(1.081) | 76.67(0.998) | 78.36(0.890) |
| THAT | 92.99(0.339) | 93.01(0.333) | 89.30(0.434) | 88.62(0.583) | 76.93(0.429) | 77.06(1.551) |
| LaxCat | 93.35(0.453) | 93.34(0.471) | 87.76(0.574) | 86.51(0.517) | 73.99(0.662) | 74.07(1.457) |
| UniTS | 93.36(0.170) | 93.28(0.191) | 85.16(1.143) | 83.65(1.121) | 74.80(0.392) | 75.91(0.615) |
| FOCAL | 93.96(0.906) | 93.94(0.923) | 87.01(0.574) | 85.27(0.213) | 71.42(0.308) | 73.84(0.847) |
| SimMMDG | 93.59(0.453) | 93.16(0.382) | 88.99(0.372) | 88.03(0.184) | 81.42(0.924) | 82.03(0.497) |
| MATE | **94.68(1.037)** | **94.72(1.004)** | **89.60(0.217)** | **88.88(0.355)** | **83.02(0.804)** | **83.96(0.209)** |

## 6 Experiments

### 6.1 Experiment Setup

**Datasets:** To evaluate the effectiveness of our method, we consider the different downstream tasks: classification, KNN evaluation, and linear probing on several multi-modal time series classification datasets. Specifically, we consider the WIFI [104], and KETI [24] datasets. Moreover, we further consider the human motion prediction datasets like Motion [82], HumanEva-I [87], H36M [34], UCIHAR [1], PAMAP2 [81], and RealWorld-HAR [89], which consider different positions of the human body as different modalities. Moreover, we also consider two healthcare datasets such as MIT-BIH [72] and D1NAMO [17], which are related to arrhythmia and

**Table 4: KNN evaluation results.**

| Model | D1NAMO | | KETI | | MIT-BIH | | UCIHAR | |
|---|---|---|---|---|---|---|---|---|
| | Accuracy | Macro-F1 | Accuracy | Macro-F1 | Accuracy | Macro-F1 | Accuracy | Macro-F1 |
| CPC | 88.81(0.721) | 52.83(0.453) | 93.89(0.121) | 79.17(0.255) | 94.91(0.306) | 93.89(0.323) | 71.54(0.959) | 71.80(0.937) |
| SimCLR | 89.76(1.360) | 56.39(1.038) | 93.81(0.512) | 79.82(0.564) | 94.48(0.221) | 93.40(0.257) | 86.33(0.550) | 86.64(0.360) |
| TS-TCC | 91.51(0.324) | 64.06(0.610) | 94.21(0.225) | 80.64(0.357) | 94.76(0.495) | 94.37(0.477) | 89.62(0.392) | 89.69(0.256) |
| COCOA | 87.42(0.535) | 47.42(0.552) | 88.79(0.669) | 67.35(0.900) | 93.72(0.523) | 91.57(0.303) | 87.41(0.586) | 87.63(0.377) |
| TS2Vec | 88.10(0.677) | 49.34(0.565) | 90.22(0.406) | 69.19(0.845) | 61.19(1.165) | 55.33(1.183) | 68.30(1.211) | 66.37(1.155) |
| Mixing-up | 89.62(0.607) | 55.30(1.084) | 91.96(0.639) | 74.75(0.450) | 96.20(0.520) | 94.02(0.274) | 91.19(0.450) | 91.24(0.474) |
| TFC | 89.30(0.283) | 52.48(0.631) | 87.30(0.901) | 66.41(1.076) | 78.13(0.386) | 74.43(0.488) | 65.91(0.421) | 65.65(0.340) |
| FOCAL | 92.02(0.130) | 67.72(0.789) | 93.93(0.448) | 79.40(0.408) | 97.96(0.066) | 97.02(0.091) | 92.49(0.386) | 92.36(0)0.282 |
| CroSSL | 90.53(0.322) | 53.87(0.560) | 90.85(0.407) | 68.79(0.545) | 96.14(0.273) | 95.47(0.287) | 91.11(0.335) | 91.09(0.338) |
| **MATE** | **92.61(0.336)** | **69.17(0.401)** | **94.79(0.555)** | **81.19(0.287)** | **98.22(0.210)** | **97.26(0.169)** | **93.48(0.388)** | **93.34(0.258)** |

noninvasive type 1 diabetes. Moreover, we have extended our consideration to encompass audio and video datasets, incorporating three multi-modality datasets—Human, Cartoon, and D2 as outlined in [9, 15], which comprise three modalities: video, audio, and pre-computed optical flow. Please refer to Appendix E for more details on the dataset descriptions.

**Evaluation Metric.** We use ADAM optimizer [42] in all experiments and report the accuracy and the Macro-F1 as evaluation metrics. Please refer to Appendix D for the implementation details.

**Baselines.** To evaluate the performance of the proposed MATE, we consider the different types of baselines. We first consider the convention ResNet [23]. Sequentially, we consider several baselines for multi-modal sensing data like STFNets [101], SenseHAR[36], THAT [51], MaCNN [79], LaxCat [25], UniTS [54], and RFNet [14]. Moreover, we also consider methods based on contrastive learning like CPC[74], SimCLR[3], MoCo[4], MTSS[83], MAE[22] CMC[90], GMC[76], TS-TCC[18], Cocoa[12], TS2Vec[105], Mixing-up[94], TFC [111], Cosmo[75], TNC[91], and CroSSL [11]. Moreover, we also consider methods that perform well in audio and video datasets, such as SimMMDG [15]. Finally, we consider the recently proposed FOCAL [64] which considers an orthogonal latent space between domain-shared and domain-specific latent variables.

## 6.2 Results and Discussion

**Time Series Classification**: Experimental results for time series classification are shown in Table 1 , 2 and 3. According to the experiment results, we can find that the proposed **MATE** model achieves the best accuracy and Macro-F1 score across different datasets. Compared with the methods based on contrastive learning and the conventional supervised learning methods, the contrastive-learning-based methods achieve better performance since they can disentangle the modality-shared and modality-specific latent variables to some extent. Moreover, since our method explicitly considers the dependence between the modality-shared and modality-specific latent variables, it outperforms the other methods like Focal and CroSSL. More interestingly, as for the experiment results of the DINAMO datasets, our method achieves a clear improvement compared with the methods with the assumption of an orthogonal latent space, which indirectly evaluates the guess mentioned in Figure 1.

**KNN Evaluation:** Following the setting of [64], we consider both the modality-shared/modality-specific latent variables and use a KNN classifier with all available labels. Experiment results are shown in Table 4 and 5. According to the experiment results, we can

**Table 5: KNN evaluation results for Realworld-HAR, and PAMAP2 datasets.**

| Model | PAMAP2 | | Realworld-HAR | |
|---|---|---|---|---|
| | Accuracy | Macro-F1 | Accuracy | Macro-F1 |
| SimCLR | 64.51(0.454) | 61.14(0.435) | 65.84(0.160) | 62.34(0.607) |
| MoCo | 69.24 | 67.66 | 74.96 | 71.34 |
| CMC | 80.32 | 79.38 | 52.16 | 58.68 |
| MAE | 68.57 | 64.27 | 87.94 | 88.17 |
| Cosmo | 80.05 | 77.43 | 81.02 | 78.17 |
| Cocoa | 71.29(0.289) | 69.74(0.244) | 77.78(0.684) | 74.59(0.573) |
| MTSS | 39.31 | 33.79 | 51.01 | 43.84 |
| TS2Vec | 56.39(1.419) | 51.80(1.695) | 64.80(0.666) | 58.32(0.590) |
| GMC | 78.43 | 75.43 | 74.15 | 75.60 |
| TNC | 79.93 | 76.53 | 78.82 | 75.65 |
| TS-TCC | 80.32(0.246) | 78.96(0.086) | 76.86(0.332) | 76.58(0.290) |
| FOCAL | 84.82(0.740) | 83.78(0.447) | 82.05(0.931) | 82.54(0.502) |
| **MATE** | **85.94(0.377)** | **84.66(0.386)** | **88.74(0.152)** | **88.92(0.090)** |

find that the proposed **MATE** still outperforms the other baselines like CroSSL. This is because the representation from our method preserves the dependencies of modality-shared and modality-specific latent variables, hence the representation contains richer semantic information and finally leads to better alignment results.

**Linear Probing:** We consider the linear probing task with four different label ratios (100%, 10%, 5%, and 1%) as shown in Table 6. The proposed MATE still consistently outperforms the state-of-the-art baselines in different label rates. Specifically, our method achieves 0.8% improvement with 100% labels, 1.9% improvement with 10% labels, 6% improvement with 5% labels, and 11% improvement with 1% labels, indirectly reflecting that MATE captures sufficient semantic information with limited labels.

## 6.3 Visualization Results

We further provide the visualization results as shown in Figure 4 to evaluate whether the proposed method can capture the semantic information effectively. We can find that our method can form better clusters with distinguished margins, meaning that the proposed method can well disentangle the latent variables. In the meanwhile, since the other methods assume the orthogonal latent space, they can not well extract the disentangled representation, and hence results in confusing clusters with unclear margins, for example, the entanglement among the "Walking", "Walking Up", and "Walking Down" in Figure 4 (b) and (e).

## 6.4 Ablation Studies

To evaluate the effectiveness of each loss term, we further devise four model variants as follows. a) **MATE-p**: we remove the KL

Table 6: Linear probing results under different label ratios on UCIHAR.

| Label Ratio | 100% | | 10% | | 5% | | 1% | |
|---|---|---|---|---|---|---|---|---|
| Model | Accuracy | Macro-F1 | Accuracy | Macro-F1 | Accuracy | Macro-F1 | Accuracy | Macro-F1 |
| CPC | 72.33(0.491) | 71.17(0.840) | 70.94(1.139) | 69.39(0.638) | 60.91(0.450) | 60.08(0.573) | 34.72(0.473) | 30.74(0.556) |
| SimCLR | 86.28(0.318) | 86.02(0.326) | 79.29(0.274) | 78.90(0.409) | 69.37(1.171) | 67.99(0.601) | 45.95(0.478) | 38.55(1.042) |
| TS-TCC | 91.40(0.220) | 91.36(0.225) | 85.50(0.434) | 84.90(0.223) | 76.62(0.307) | 74.93(0.366) | 60.52(0.591) | 58.37(0.193) |
| COCOA | 91.76(0.465) | 91.91(0.508) | 67.31(0.451) | 66.90(0.613) | 54.30(0.575) | 54.10(0.757) | 33.57(0.340) | 32.94(0.209) |
| TS2Vec | 70.78(0.212) | 68.70(0.247) | 63.88(1.280) | 61.97(1.399) | 62.46(0.087) | 60.36(0.133) | 49.31(0.148) | 42.44(0.229) |
| Mixing-up | 90.29(0.253) | 90.17(0.283) | 85.52(0.663) | 85.16(0.655) | 78.00(0.573) | 77.03(0.821) | 33.51(0.338) | 20.51(0.700) |
| TFC | 66.01(0.338) | 65.38(0.079) | 53.86(0.430) | 46.14(0.711) | 45.23(0.556) | 44.15(0.230) | 40.94(0.222) | 39.00(0.274) |
| FOCAL | 93.01(0.057) | 92.87(0.040) | 89.30(0.307) | 89.03(0.347) | 80.55(0.401) | 79.62(0.405) | 67.48(0.405) | 63.30(0.126) |
| CroSSL | 92.80(0.057) | 92.86(0.050) | 87.38(0.421) | 86.57(1.109) | 77.50(0.524) | 76.72(0.319) | 48.58(1.004) | 47.57(0.415) |
| MATE | **93.76(0.057)** | **93.66(0.025)** | **91.02(0.166)** | **90.95(0.136)** | **84.49(0.216)** | **84.54(0.323)** | **74.68(0.639)** | **69.61(0.972)** |

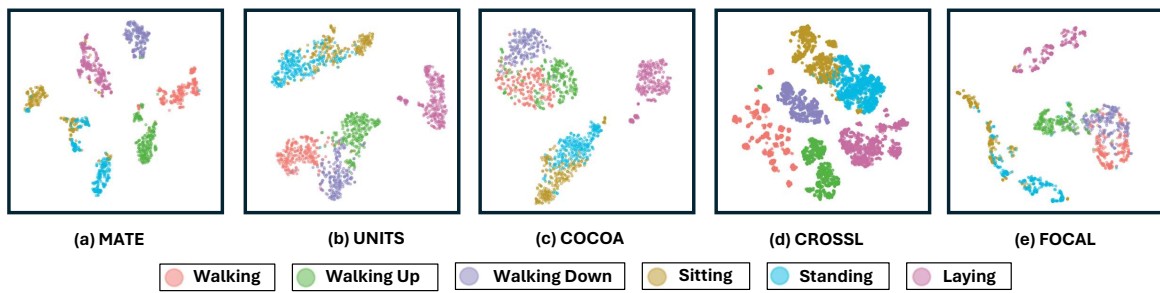

(a) MATE (b) UNITS (c) COCOA (d) CROSSL (e) FOCAL

● Walking ● Walking Up ● Walking Down ● Sitting ● Standing ● Laying

Figure 4: The t-SNE visualization of the extracted domain-shared latent variables.

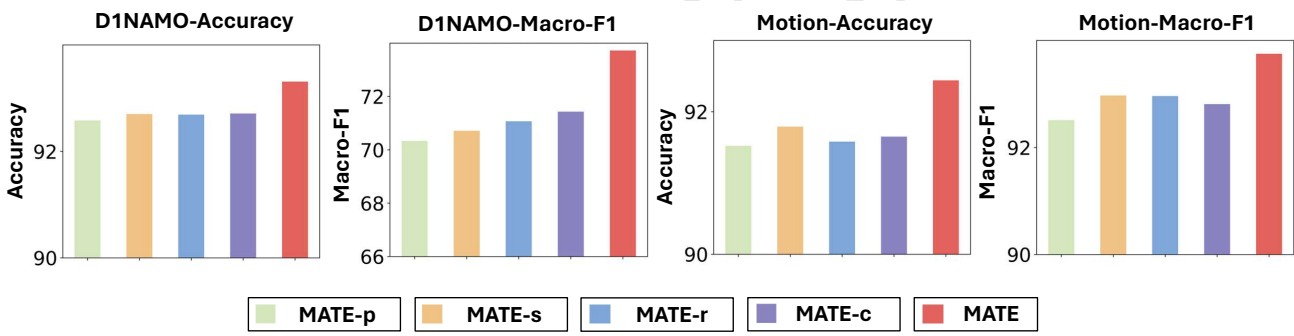

■ MATE-p ■ MATE-s ■ MATE-r ■ MATE-c ■ MATE

Figure 5: Ablation study on the DINAMO and the Motion datasets.

divergence terms for domain-specific latent variables. b) **MATE-s**: we remove the KL divergence terms for domain-shared latent variables. c) **MATE-r**: We remove the reconstruction loss. d) **MATE-c**: We remove the modality-shared constraint. Experiment results of the ablation studies on the D1NAMO and Motion datasets are shown in Figure 5. We can draw the following conclusions 1) all the loss terms play an important role in the representation learning. 2) In the D1NAMO dataset, by removing the KL divergence terms for domain-shared and domain-specific latent variables, the model performance drops, showing that these loss terms benefit the identifiability of latent variables under dependence latent space. 3) Moreover, the drop in the performance of **MATE-r** and **MATE-c** reflects that the reconstruction loss and the modality-shared constraint are conducive to preserving the semantic information.

## 6.5 Sensitivity Analysis

We also perform a sensitivity test on the loss weight values on the Motion dataset, as shown in Figure 6. We try different values of hyperparameters of $\alpha$, $\beta$, and $\gamma$. According to the experiment results, we find that 1) $L_r$ plays an important role when the values of $\alpha$

are 1 ad $1e-4$ for D1NAMO and Motion datasets, respectively. 2) our method achieves the best results when the values of $\beta$ are from $1e-3$ to $1e-2$, showing that the KL divergence terms have stable influences. 3) we also find that our method can achieve good results when the value of $\gamma$ is around $1e-2$.

## 7 Conclusion

We propose a representation learning framework for multi-modal time series data with theoretical guarantees, which breakthroughs the conventional orthogonal latent space assumption. Based on the data generation process for multi-modal time series data with dependent latent subspace, we devise a general disentangled representation learning framework with identifiability guarantees. Compared with the existing methods, the proposed **MATE** model can learn the disentangled time series representations even in the dependent latent subspace, hence our method is closer to the real-world scenarios. Evaluation on the time series classification, KNN evaluation, and linear probing on several multi-modal time series datasets illustrate the effectiveness of our method.

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

# A Proof of Modality-shared Latent Variables $z_t^c$

## A.1 Proof of Subspace Identification

THEOREM 1. (**Subspace Identification of the Modality-shared and Modality-specific Latent Variables**) *Suppose that the observed data from different modalities is generated following the data generation process in Figure 2, and we further make the following assumptions:*

- *A1 (Smooth and Positive Density:) The probability density of latent variables is smooth and positive, i.e., $p(z_t|z_{t-1}) > 0$ over $\mathcal{Z}_t$ and $\mathcal{Z}_{t-1}$.*
- *A2 (Conditional Independence:) Conditioned on $z_{t-1}$, each $z_{t,i}^c$ is independent of $z_{t,j}^c$ for $i, j \in \{1, \cdots, n_c\}, i \neq j$. And conditioned on $z_{t-1}$ and $z_t^c$, each $z_{t,i}^{s_m}$ is independent of $z_{t,j}^{s_m}$, for $i, j \in \{n_c + 1, \cdots, n\}, i \neq j$.*
- *A3 (non-singular Jacobian): Each $g_m$ has non-singular Jacobian matrices almost anywhere and $g_m$ is invertible.*
- *A4 (Linear Independence:) For any $z_t^{s_*} \in \mathcal{Z}_t^{s_*}$, there exist $n_c + 1$ values of $z_{t-1,k}^{s_m}, k = n_c + 1, \cdots, n$, such that these vectors $v_{t,j}$ are linearly independent, where $v_{t,j}$ are defined as follows:*

$$v_{t,j} = \left( \frac{\partial^2 \log p(z_{t,j}^{s_m}|z_{t-1}^m, z_t^c)}{\partial z_{t,j}^{s_m} \partial z_{t-1,n_c+1}^{s_m}}, \cdots, \frac{\partial^2 \log p(z_{t,j}^{s_m}|z_{t-1}^m, z_t^c)}{\partial z_{t,j}^{s_m} \partial z_{t-1,n}^{s_m}} \right) \quad (17)$$

*Then if $\hat{g}_1 : \mathcal{Z}_t^c \times \mathcal{Z}_t^{s_1} \to \mathcal{X}_t^{s_1}$ and $\hat{g}_2 : \mathcal{Z}_t^c \times \mathcal{Z}_t^{s_2} \to \mathcal{X}_t^{s_2}$ assume the generating process of the true model $(g_1, g_2)$ and match the joint distribution $p(x_t^{s_1}, x_t^{s_2})$ of each time step then $z_t^c$ is subspace identifiable.*

PROOF. For $(x_t^1, x_t^2) \sim p(x_t^1, x_t^2)$, because of the matched joint distribution, we have the following relations between the true variables $z_t^c, z_t^{s_1}, z_t^{s_2}$ and the estimated ones $\hat{z}_t^c, \hat{z}_t^{s_1}, \hat{z}_t^{s_2}$:

$$x_t^{s_1} = g_1(z_t^c, z_t^{s_1}) = \hat{g}_1(\hat{z}_t^c, \hat{z}_t^{s_1}) \quad (18)$$

$$x_t^{s_2} = g_2(z_t^c, z_t^{s_2}) = \hat{g}_2(\hat{z}_t^c, \hat{z}_t^{s_2}) \quad (19)$$

$$(\hat{z}_t^c, \hat{z}_t^{s_1}, \hat{z}_t^{s_2}) = \hat{g}^{-1}(x_t^1, x_t^2) = \hat{g}^{-1}(g(z_t^c, z_t^{s_1}, z_t^{s_2})) := h(z_t^c, z_t^{s_1}, z_t^{s_2}), \quad (20)$$

where $\hat{g}_1, \hat{g}_2$ are the estimated invertible generating function and $h := \hat{g}^{-1} \circ g$ denotes a smooth and invertible function that transforms the true variables $z_t^c, z_t^{s_1}, z_t^{s_2}$ to the estimated ones $\hat{z}_t^c, \hat{z}_t^{s_1}, \hat{z}_t^{s_2}$.

By combining Equation (20) and (18), we have

$$g_1(z_t^c, z_t^{s_1}) = \hat{g}_1(h_{c,s_1}(z_t^c, z_t^{s_1}, z_t^{s_2})). \quad (21)$$

For $i \in \{1, \cdots, n_{x^{s_1}}\}$ and $j \in \{1, \cdots, n_{s_2}\}$, we take a partial derivative of the $i$-th dimension of $x_t^{s_1}$ on both sides of Equation (21) w.r.t. $z_{t,j}^{s_2}$ and have:

$$0 = \frac{\partial g_{1,i}(z_t^c, z_t^{s_1})}{\partial z_{t,j}^{s_2}} = \frac{\partial \hat{g}_{1,i}(h_{c,s_1}(z_t^c, z_t^{s_1}))}{\partial z_{t,j}^{s_2}}. \quad (22)$$

The aforementioned equation equals 0 because there is no $z_{t,j}^{s_2}$ in the left-hand side of the equation. By expanding the derivative on the right-hand side, we further have:

$$\sum_{k \in \{1, \cdots, n_c+n_{s_1}\}} \frac{\partial \hat{g}_{1,i}(z_t^c, z_t^{s_1})}{\partial h_{(c,s_1),k}} \cdot \frac{\partial h_{(c,s_1),k}(z_t^c, z_t^{s_1}, z_t^{s_2})}{\partial z_{t,j}^{s_2}} = 0. \quad (23)$$

Since $\hat{g}_1$ is invertible, the determinant of $J_{\hat{g}_1}$ does not equal to 0, meaning that for $n_c + n_{s_1}$ different values of $\hat{g}_{1,i}$, each vector $\left[ \frac{\partial \hat{g}_{1,i}(z_t^c, z_t^{s_1})}{\partial h_{(c,s_1),1}}, \cdots, \frac{\partial \hat{g}_{1,i}(z_t^c, z_t^{s_1})}{\partial h_{(c,s_1),n_c+n_{s_1}}} \right]$ are linearly independent. Therefore, the $(n_c + n_{s_1}) \times (n_c + n_{s_1})$ linear system is invertible and has the unique solution as follows:

$$\frac{\partial h_{(c,s_1),k}(z_t^c, z_t^{s_1}, z_t^{s_2})}{\partial z_{t,j}^{s_2}} = 0. \quad (24)$$

According to Equation (24), for any $k \in \{1, \cdots, n_c + n_{s_1}\}$ and $j \in \{1, \cdots, n_{s_2}\}$, $h_{(c,s_1),k}(z_t^c, z_t^{s_1}, z_t^{s_2})$ does not depend on $z_t^{s_2}$. In other word, $\{z_t^c, z_t^{s_1}\}$ does not depend on $z_t^{s_2}$.

Similarly, by combining Equation (20) and (19), we have

$$g_2(z_t^c, z_t^{s_2}) = \hat{g}_2(h_{c,s_2}(z_t^c, z_t^{s_1}, z_t^{s_2})). \quad (25)$$

For $i \in \{1, \cdots, n_{x^{s_2}}\}$ and $j \in \{1, \cdots, n_{s_1}\}$, we take a partial derivative of the $i$-th dimension of $x_t^{s_2}$ on both sides of Equation (25) w.r.t $z_{t,j}^{s_1}$ and have:

$$
\begin{aligned}
0 = \frac{\partial g_{2,i}(z_t^c, z_t^{s_2})}{\partial z_{t,j}^{s_1}} &= \frac{\partial \hat{g}_{2,i}(h_{c,s_2}(z_t^c, z_t^{s_2}))}{\partial z_{t,j}^{s_1}} \\
&= \sum_{k \in \{1 \cdots, n_c+n_{s_2}\}} \frac{\partial \hat{g}_{2,i}(z_t^c, z_t^{s_2})}{\partial h_{(c,s_2),k}} \cdot \frac{\partial h_{(c,s_2),k}(z_t^c, z_t^{s_1}, z_t^{s_2})}{\partial z_{t,j}^{s_1}}
\end{aligned}
\quad (26)
$$

Since $\hat{g}_2$ is invertible, for $n_c + n_{s_2}$ different values of $\hat{g}_{2,i}$, each vector $\left[ \frac{\partial \hat{g}_{2,i}(z_t^c, z_t^{s_2})}{\partial h_{(c,s_2),1}}, \cdots, \frac{\partial \hat{g}_{2,i}(z_t^c, z_t^{s_2})}{\partial h_{(c,s_2),n_c+n_{s_2}}} \right]$ are linearly independent. Therefore, the $(n_c + n_{s_2}) \times (n_c + n_{s_2})$ linear system is invertible and has the unique solution as follows:

$$\frac{\partial h_{(c,s_2),k}(z_t^c, z_t^{s_1}, z_t^{s_2})}{\partial z_{t,j}^{s_1}} = 0, \quad (27)$$

meaning that $\{z_t^c, z_t^{s_2}\}$ does not depend on $z_t^{s_1}$.

According to Equation (20), we have $\hat{z}_t^c = h_c(z_t^c, z_t^{s_1}, z_t^{s_2})$. By using the fact that $\{z_t^c, z_t^{s_2}\}$ does not depend on $z_t^{s_1}$ and $\{z_t^c, z_t^{s_1}\}$ does not depend on $z_t^{s_2}$, we have $\hat{z}_t^c = h_c(z_t^c)$, i.e., the modality-shared latent variables are subspace identifiable.

Since the matched marginal distribution of $p(x_t^{s_1}|x_{t-1}^{s_1})$, we have:

$$
\begin{aligned}
&\forall x_{t-1}^{s_1} \in \mathcal{X}_{t-1}^{s_1}, \\
&p(\hat{x}_t^{s_1}|x_{t-1}^{s_1}) = p(x_t^{s_1}|x_{t-1}^{s_1}) \\
&\iff p(\hat{g}_1(\hat{z}_t^1)|x_{t-1}^{s_1}) = p(g_1(z_t^1)|x_{t-1}^{s_1}),
\end{aligned}
\quad (28)
$$

where $z_t^1 = \{z_t^c, z_t^{s_1}\}$ and $\hat{z}_t^1 = \{\hat{z}_t^c, \hat{z}_t^{s_1}\}$. Sequentially, by using the change of variables formula, we can further obtain Equation (29)

$$
\begin{aligned}
&p(\hat{g}_1(\hat{z}_t^1)|x_{t-1}^{s_1}) = p(g_1(z_t^1)|x_{t-1}^{s_1}) \\
&\iff p(g_1^{-1} \circ \hat{g}_1(\hat{z}_t^1)|x_{t-1}^{s_1})|J_{g_1^{-1}}| = p(z_t^1|x_{t-1}^{s_1})|J_{g_1^{-1}}| \\
&\iff p(h_1(\hat{z}_t^1)|x_{t-1}^{s_1}) = p(z_t^1|x_{t-1}^{s_1}) \\
&\iff p(h_1(\hat{z}_t^1)|\hat{z}_{t-1}^1) = p(z_t^1|z_{t-1}^1),
\end{aligned}
\quad (29)
$$

where $h_1 := g_1^{-1} \circ \hat{g}_1$ is the transformation between the ground-true and the estimated latent variables. $J_{g_1^{-1}}$ denotes the absolute value of Jacobian matrix determinant of $g_1^{-1}$. Since we assume that $g_1$ and $\hat{g}_1$ are invertible, $|J_{g^{-1}}| \neq 0$ and $h_1$ is also invertible.

According to the A2 (conditional independent assumption), we can have Equation (30)

$$p(z_t^1|z_{t-1}^1) = \prod_{i=1}^{n} p(z_{t,i}^1|z_{t-1}^1); \quad p(\hat{z}_t^1|\hat{z}_{t-1}^1) = \prod_{i=1}^{n} p(\hat{z}_{t,i}^1|\hat{z}_{t-1}^1).$$

(30)

For convenience, we take logarithm on both sides of Equation (30) and have:

$$\log p(z_t^1|z_{t-1}^1) = \sum_{i=1}^{n} \log p(z_{t,i}^1|z_{t-1}^1);$$

$$\log p(\hat{z}_t^1|\hat{z}_{t-1}^1) = \sum_{i=1}^{n} \log p(\hat{z}_{t,i}^1|\hat{z}_{t-1}^1).$$

(31)

By combining Equation (31) and Equation (29), we have:

$$p(h_1(\hat{z}_t^1)|\hat{z}_{t-1}^1) = p(z_t^1|z_{t-1}^1) \iff p(\hat{z}_t^1|\hat{z}_{t-1}^1)|\mathbf{J}_{h^{-1}}| = p(z_t^1|z_{t-1}^1)$$

$$\iff \sum_{i=1}^{n} \log p(\hat{z}_{t,i}^1|\hat{z}_{t-1}^1)$$

$$= \sum_{i=1}^{n} \log p(z_{t,i}^1|z_{t-1}^1) - \log|\mathbf{J}_{h^{-1}}|,$$

(32)

where $\mathbf{J}_{h^{-1}}$ are the Jacobian matrix of $h^{-1}$.

Sequentially, we take the first-order derivative with $\hat{z}_{t,i}^c$, where $i \in \{1, \cdots, n_c\}$ and have:

$$\frac{\partial \log p(\hat{z}_t^1|\hat{z}_{t-1}^1)}{\partial \hat{z}_{t,i}^c}$$

$$= \sum_{j=1}^{n_c} \frac{\partial \log p(\hat{z}_{t,j}^c|\hat{z}_{t-1}^1)}{\partial \hat{z}_{t,i}^c} + \sum_{j=n_c+1}^{n} \frac{\partial \log p(\hat{z}_{t,j}^{s_1}|\hat{z}_{t-1}^1, \hat{z}_t^c)}{\partial \hat{z}_{t,i}^c}$$

$$= \sum_{j=1}^{n_c} \frac{\partial \log p(z_{t,j}^c|z_{t-1}^1)}{\partial z_{t,j}^c} \cdot \frac{\partial z_{t,j}^c}{\partial \hat{z}_{t,i}^c}$$

$$+ \sum_{j=n_c+1}^{n} \frac{\partial \log p(z_{t,j}^{s_1}|z_{t-1}^1, z_t^c)}{\partial z_{t,j}^{s_1}} \cdot \frac{\partial z_{t,j}^{s_1}}{\partial \hat{z}_{t,i}^c} - \frac{\partial |\mathbf{J}_{h^{-1}}|}{\partial \hat{z}_{t,i}^c}.$$

(33)

Then we further take the second-order derivative w.r.t $z_{t-1,k}^{s_1}$, where $k \in \{n_c + 1, \cdots, n\}$ and we have:

$$\sum_{j=1}^{n_c} \frac{\partial^2 \log p(\hat{z}_{t,j}^c|\hat{z}_{t-1}^1)}{\partial \hat{z}_{t,i}^c \partial z_{t-1,k}^{s_1}} + \sum_{j=n_c+1}^{n} \frac{\partial^2 \log p(\hat{z}_{t,j}^{s_1}|\hat{z}_{t-1}^1, \hat{z}_t^c)}{\partial \hat{z}_{t,i}^c \partial z_{t-1,k}^{s_1}}$$

$$= \sum_{j=1}^{n_c} \frac{\partial^2 \log p(z_{t,j}^c|z_{t-1}^1)}{\partial z_{t,j}^c \partial z_{t-1,k}^{s_1}} \cdot \frac{\partial z_{t,j}^c}{\partial \hat{z}_{t,i}^c}$$

$$+ \sum_{j=n_c+1}^{n} \frac{\partial^2 \log p(z_{t,j}^{s_1}|z_{t-1}^1, z_t^c)}{\partial z_{t,j}^{s_1} \partial z_{t-1,k}^{s_1}} \cdot \frac{\partial z_{t,j}^{s_1}}{\partial \hat{z}_{t,i}^c} - \frac{\partial^2 |\mathbf{J}_{h^{-1}}|}{\partial \hat{z}_{t,i}^c \partial z_{t-1,k}^{s_1}}.$$

(34)

Since $\hat{z}_{t,j}^c$ does not change across different values of $z_{t-1,k}^{s_1}$, then $\frac{\partial^2 \log p(\hat{z}_{t,j}^c|\hat{z}_{t-1}^1)}{\partial \hat{z}_{t,i}^c \partial z_{t-1,k}^{s_1}} = 0$. Since $\frac{\partial^2 \log p(\hat{z}_{t,j}^{s_1}|\hat{z}_{t-1}^1, \hat{z}_t^c)}{\partial \hat{z}_{t,i}^c}$ does not change across different values of $z_{t-1,k}^{s_1}$, then $\frac{\partial^2 \log p(\hat{z}_{t,j}^{s_1}|\hat{z}_{t-1}^1, \hat{z}_t^c)}{\partial \hat{z}_{t,i}^c \partial z_{t-1,k}^{s_1}} = 0$. Moreover, since $\frac{\partial^2 \log p(z_{t,j}^c|z_{t-1}^1)}{\partial z_{t,j}^c \partial z_{t-1,k}^{s_1}}$ and $\frac{\partial^2 |\mathbf{J}_{h^{-1}}|}{\partial \hat{z}_{t,i}^c \partial z_{t-1,k}^{s_1}} = 0$, Equation (34) can

be further rewritten as:

$$\sum_{j=n_c+1}^{n} \frac{\partial^2 \log p(z_{t,j}^{s_1}|z_{t-1}^1, z_t^c)}{\partial z_{t,j}^{s_1} \partial z_{t-1,k}^{s_1}} \cdot \frac{\partial z_{t,j}^{s_1}}{\partial \hat{z}_{t,i}^c} = 0.$$

(35)

By leveraging the linear independence assumption, the linear system denoted by Equation (35) has the only solution $\frac{\partial z_{t,j}^{s_1}}{\partial \hat{z}_{t,i}^c} = 0$. As $h_1$ is smooth, its Jacobian can written as:

$$\mathbf{J}_{h_1} = \left[ \begin{array}{c|c} \mathbf{A} := \frac{\partial z_t^c}{\partial \hat{z}_t^c} & \mathbf{B} := \frac{\partial z_t^c}{\partial \hat{z}_t^{s_1}} = 0 \\ \hline \mathbf{C} := \frac{\partial z_t^{s_1}}{\partial \hat{z}_t^c} = 0 & \mathbf{D} := \frac{\partial z_t^{s_1}}{\partial \hat{z}_t^{s_1}}. \end{array} \right]$$

(36)

Therefore, $z_t^{s_1}$ is subspace identifiable. Similarly, we can prove that $z_t^{s_m}$ is subspace identifiable. □

## A.2 Proof of Component-wise Identification

COROLLARY 1.1. (***Component-wise Identification of the Modality-shared and Modality-specific Latent Variables***) *Suppose that the observed data from different modalities is generated following the data generation process in Figure 2, and we further make the following assumptions:*

- *A1 (Smooth and Positive Density:) The probability density of latent variables is smooth and positive, i.e., $p(z_t|z_{t-1}) > 0$ over $\mathcal{Z}_t$ and $\mathcal{Z}_{t-1}$.*
- *A2 (Conditional Independence:) Conditioned on $z_{t-1}$, each $z_{t,i}^c$ is independent of $z_{t,j}^c$ for $i, j \in \{1, \cdots, n_c\}, i \neq j$. And conditioned on $z_{t-1}$ and $z_t^c$, each $z_{t,i}^{s_m}$ is independent of $z_{t,j}^{s_m}$, for $i, j \in \{n_c + 1, \cdots, n\}, i \neq j$.*
- *A3 (Linear Independence:) For any $z_t \in \mathcal{Z}_t$, there exist $2n+1$ values of $z_{t-1,k}^m, k = 1, \cdots, n$, such that these vectors $v_{t,l}$ are linearly independent, where $v_{t,l}$ are defined as follows:*

$$v_{t,l} = \Big( \frac{\partial^3 \log p(z_{t,l}^c|z_{t-1}^m)}{\partial^2 z_{t,l}^c \partial z_{t-1,1}^m}, \cdots, \frac{\partial^3 \log p(z_{t,l}^c|z_{t-1}^m)}{\partial^2 z_{t,l}^c \partial z_{t-1,n}^m},$$

$$\frac{\partial^2 \log p(z_{t,l}^c|z_{t-1}^m)}{\partial z_{t,l}^c \partial z_{t-1,1}^m}, \cdots, \frac{\partial^2 \log p(z_{t,l}^c|z_{t-1}^m)}{\partial z_{t,l}^c \partial z_{t-1,n}^m},$$

$$\frac{\partial^3 \log p(z_{t,l}^{s_m}|z_{t-1}^m, z_t^c)}{\partial^2 z_{t,l}^{s_m} \partial z_{t-1,1}^m}, \cdots, \frac{\partial^3 \log p(z_{t,l}^{s_m}|z_{t-1}^m, z_t^c)}{\partial^2 z_{t,l}^{s_m} \partial z_{t-1,n}^m},$$

$$\frac{\partial^2 \log p(z_{t,l}^{s_m}|z_{t-1}^m, z_t^c)}{\partial z_{t,l}^{s_m} \partial z_{t-1,1}^m}, \cdots, \frac{\partial^2 \log p(z_{t,l}^{s_m}|z_{t-1}^m, z_t^c)}{\partial z_{t,l}^{s_m} \partial z_{t-1,n}^m} \Big)$$

(37)

*Then if $\hat{g}_1 : \mathcal{Z}_t^c \times \mathcal{Z}_t^{s_1} \to \mathcal{X}_t^{s_1}$ and $\hat{g}_2 : \mathcal{Z}_t^c \times \mathcal{Z}_t^{s_2} \to \mathcal{X}_t^{s_2}$ assume the generating process of the true model $(g_1, g_2)$ and match the joint distribution $p(x_t^{s_1}, x_t^{s_2})$ of each time step then $z_t^c$ is component-wise identifiable.*

PROOF. Then we let $z_t^1 = \{z_t^c, z_t^{s_1}\}$ and $\hat{z}_t^1 = \{\hat{z}_t^c, \hat{z}_t^{s_1}\}$. According to Equation (2), we have $\hat{z}_t = h_1(z_t)$, where $h_1 := \hat{g}_1^{-1} \circ g_1$ is an invertible function. Sequentially, it is straightforward to see that if the components of $\hat{z}_t^{s_1}$ are mutually independent conditional on $\hat{z}_{t-1}^{s_1}$ and $\hat{z}_t^c$, the components of $\hat{z}_t^c$ are mutually independent conditional on $\hat{z}_{t-1}^c$, then for any $i \neq j$, we have:

$$\frac{\partial^2 \log p(\hat{z}_t^{s_1}|\hat{z}_{t-1}^{s_1}, \hat{z}_t^c)}{\partial \hat{z}_{t,i}^{s_1} \partial \hat{z}_{t,j}^{s_1}} = 0, \frac{\partial^2 \log p(\hat{z}_t^c|\hat{z}_{t-1}^c)}{\partial \hat{z}_{t,i}^c \partial \hat{z}_{t,j}^c} = 0,$$

(38)

by assuming that the second-order derivative exists. The Jacobian matrix of the mapping from $(x_{t-1}^{s_1}, z_t^1)$ to $(x_{t-1}^{s_1}, \hat{z}_t^1)$ is $\begin{bmatrix} \mathbb{I} & 0 \\ * & H_t^{s_1} \end{bmatrix}$, where $H_t^{s_1}$ denotes the absolute value of the determinant of this Jacobian matrix is $|H_t^{s_1}|$. Therefore, $p(\hat{z}_t^1, x_{t-1}^{s_1}) \cdot |H_t^{s_1}| = p(z_t^1, x_{t-1}^{s_1})$. Dividing both sides of this equation by $p(x_{t-1}^{s_1})$ gives

$$p(\hat{z}_t^1 | x_{t-1}^{s_1}) \cdot |H_t^{s_1}| = p(z_t^1 | x_{t-1}^{s_1}). \tag{39}$$

Since $p(z_t^1 | z_{t-1}^1) = p(z_t^1 | g_1(z_{t-1}^1)) = p(z_t^1 | x_{t-1}^{s_1})$ and similarly $p(\hat{z}_t^1 | \hat{z}_{t-1}^1) = p(\hat{z}_t^1 | x_{t-1}^{s_1})$, so we further have:

$$\log p(\hat{z}_t^1 | \hat{z}_{t-1}^1) = \log p(z_t^1 | z_{t-1}^1) - \log |H_t^{s_1}|. \tag{40}$$

According to Equation (40), we take the first-order derivative with $\hat{z}_{t,i}^c$, where $i \in \{1, \cdots, n_c\}$ and have:

$$\begin{aligned}
&\frac{\partial \log p(\hat{z}_t^1 | \hat{z}_{t-1}^1)}{\partial \hat{z}_{t,i}^c} \\
&= \sum_{l=1}^{n_c} \frac{\partial \log p(\hat{z}_{t,l}^c | \hat{z}_{t-1}^1)}{\partial \hat{z}_{t,i}^c} + \sum_{l=n_c+1}^{n} \frac{\partial \log p(\hat{z}_{t,l}^{s_1} | \hat{z}_{t-1}^1, \hat{z}_t^c)}{\partial \hat{z}_{t,i}^c} \\
&= \sum_{l=1}^{n_c} \frac{\partial \log p(z_{t,l}^c | z_{t-1}^1)}{\partial z_{t,l}^c} \cdot \frac{\partial z_{t,l}^c}{\partial \hat{z}_{t,i}^c} \\
&+ \sum_{l=n_c+1}^{n} \frac{\partial \log p(z_{t,l}^{s_1} | z_{t-1}^1, z_t^c)}{\partial z_{t,l}^{s_1}} \cdot \frac{\partial z_{t,l}^{s_1}}{\partial \hat{z}_{t,i}^c} - \frac{\partial \log |H_t^{s_1}|}{\partial \hat{z}_{t,i}^c}.
\end{aligned} \tag{41}$$

Then we further take the second-order derivative w.r.t $\hat{z}_{t,j}^c$, where $j \in \{1, \cdots, n_c\}$ and we have:

$$\begin{aligned}
&\sum_{l=1}^{n_c} \frac{\partial^2 \log p(\hat{z}_{t,l}^c | \hat{z}_{t-1}^1)}{\partial \hat{z}_{t,i}^c \partial \hat{z}_{t,j}^c} + \sum_{l=n_c+1}^{n} \frac{\partial^2 \log p(\hat{z}_{t,l}^{s_1} | \hat{z}_{t-1}^1, \hat{z}_t^c)}{\partial \hat{z}_{t,i}^c \partial \hat{z}_{t,j}^c} \\
&= \sum_{l=1}^{n_c} \frac{\partial^2 \log p(z_{t,l}^c | z_{t-1}^1)}{\partial^2 z_{t,l}^c} \cdot \frac{\partial z_{t,l}^c}{\partial \hat{z}_{t,j}^c} \cdot \frac{\partial z_{t,l}^c}{\partial \hat{z}_{t,i}^c} \\
&+ \sum_{l=1}^{n_c} \frac{\partial \log p(z_{t,l}^c | z_{t-1}^1)}{\partial z_{t,l}^c} \cdot \frac{\partial^2 z_{t,l}^c}{\partial \hat{z}_{t,i}^c \partial \hat{z}_{t,j}^c} \\
&+ \sum_{l=n_c+1}^{n} \frac{\partial^2 \log p(z_{t,l}^{s_1} | z_{t-1}^1, z_t^c)}{\partial^2 z_{t,l}^{s_1}} \cdot \frac{\partial z_{t,l}^{s_1}}{\partial \hat{z}_{t,j}^c} \cdot \frac{\partial z_{t,l}^{s_1}}{\partial \hat{z}_{t,i}^c} \\
&+ \sum_{l=n_c+1}^{n} \frac{\partial \log p(z_{t,l}^{s_1} | z_{t-1}^1, z_t^c)}{\partial z_{t,l}^{s_1}} \cdot \frac{\partial^2 z_{t,l}^{s_1}}{\partial \hat{z}_{t,i}^c \partial \hat{z}_{t,j}^c} - \frac{\partial^2 \log |H_t^{s_1}|}{\partial \hat{z}_{t,i}^c \partial \hat{z}_{t,j}^c}.
\end{aligned} \tag{42}$$

Sequentially, for $k = 1, \cdots, n_c$, and each value $z_{t-1,k}^c$, the third-order derivative w.r.t. $v_{t-1,k}^c$, and we have:

$$\begin{aligned}
&\sum_{l=1}^{n_c} \frac{\partial^3 \log p(\hat{z}_{t,l}^c | \hat{z}_{t-1}^1)}{\partial \hat{z}_{t,i}^c \partial \hat{z}_{t,j}^c \partial z_{t-1,k}^c} + \sum_{l=n_c+1}^{n} \frac{\partial^3 \log p(\hat{z}_{t,l}^{s_1} | \hat{z}_{t-1}^1, \hat{z}_t^c)}{\partial \hat{z}_{t,i}^c \partial \hat{z}_{t,j}^c \partial z_{t-1,k}^c} \\
&= \sum_{l=1}^{n_c} \frac{\partial^3 \log p(z_{t,l}^c | z_{t-1}^1)}{\partial^2 z_{t,l}^c \partial z_{t-1,k}^c} \cdot \frac{\partial z_{t,l}^c}{\partial \hat{z}_{t,j}^c} \cdot \frac{\partial z_{t,l}^c}{\partial \hat{z}_{t,i}^c} \\
&+ \sum_{l=1}^{n_c} \frac{\partial^2 \log p(z_{t,l}^c | z_{t-1}^1)}{\partial z_{t,l}^c \partial z_{t-1,k}^c} \cdot \frac{\partial^2 z_{t,l}^c}{\partial \hat{z}_{t,i}^c \partial \hat{z}_{t,j}^c} \\
&+ \sum_{l=n_c+1}^{n} \frac{\partial^3 \log p(z_{t,l}^{s_1} | z_{t-1}^1, z_t^c)}{\partial^2 z_{t,l}^{s_1} \partial z_{t-1,k}^c} \cdot \frac{\partial z_{t,l}^{s_1}}{\partial \hat{z}_{t,j}^c} \cdot \frac{\partial z_{t,l}^{s_1}}{\partial \hat{z}_{t,i}^c} \\
&+ \sum_{l=n_c+1}^{n} \frac{\partial^2 \log p(z_{t,l}^{s_1} | z_{t-1}^1, z_t^c)}{\partial z_{t,l}^{s_1} \partial z_{t-1,k}^c} \cdot \frac{\partial^2 z_{t,l}^{s_1}}{\partial \hat{z}_{t,i}^c \partial \hat{z}_{t,j}^c} - \frac{\partial^3 \log |H_t^{s_1}|}{\partial \hat{z}_{t,i}^c \partial \hat{z}_{t,j}^c \partial z_{t-1,k}^c}.
\end{aligned} \tag{43}$$

Since according to Equation(38),then $\frac{\partial^3 \log p(\hat{z}_{t,l}^c | z_{t-1}^1)}{\partial \hat{z}_{t,i}^c \partial \hat{z}_{t,j}^c \partial z_{t-1,k}^c} = 0$. Since $\hat{z}_{t,l}^{s_1}$ does not change across different values of $z_{t-1,k}^c$, then $\frac{\partial^3 \log p(\hat{z}_{t,l}^{s_1} | \hat{z}_{t-1}^1, \hat{z}_t^c)}{\partial \hat{z}_{t,i}^c \partial \hat{z}_{t,j}^c \partial z_{t-1,k}^c} = 0$. Equation (43) can be further rewritten as:

$$\begin{aligned}
&\sum_{l=1}^{n_c} \frac{\partial^3 \log p(z_{t,l}^c | z_{t-1}^1)}{\partial^2 z_{t,l}^c \partial z_{t-1,k}^c} \cdot \frac{\partial z_{t,l}^c}{\partial \hat{z}_{t,j}^c} \cdot \frac{\partial z_{t,l}^c}{\partial \hat{z}_{t,i}^c} \\
&+ \sum_{l=1}^{n_c} \frac{\partial^2 \log p(z_{t,l}^c | z_{t-1}^1)}{\partial z_{t,l}^c \partial z_{t-1,k}^c} \cdot \frac{\partial^2 z_{t,l}^c}{\partial \hat{z}_{t,i}^c \partial \hat{z}_{t,j}^c} \\
&+ \sum_{l=n_c+1}^{n} \frac{\partial^3 \log p(z_{t,l}^{s_1} | z_{t-1}^1, z_t^c)}{\partial^2 z_{t,l}^{s_1} \partial z_{t-1,k}^c} \cdot \frac{\partial z_{t,l}^{s_1}}{\partial \hat{z}_{t,j}^c} \cdot \frac{\partial z_{t,l}^{s_1}}{\partial \hat{z}_{t,i}^c} \\
&+ \sum_{l=n_c+1}^{n} \frac{\partial^2 \log p(z_{t,l}^{s_1} | z_{t-1}^1, z_t^c)}{\partial z_{t,l}^{s_1} \partial z_{t-1,k}^c} \cdot \frac{\partial^2 z_{t,l}^{s_1}}{\partial \hat{z}_{t,i}^c \partial \hat{z}_{t,j}^c} = 0.
\end{aligned} \tag{44}$$

where we have made use of the fact that entries of $H_t^{s_1}$ do not depend on $z_{t-1,l}^c$. Then by leveraging the linear independence assumption, the linear system denoted by Equation (44) has the only solution $\frac{\partial z_{t,l}^c}{\partial \hat{z}_{t,i}^c} \cdot \frac{\partial z_{t,l}^c}{\partial \hat{z}_{t,j}^c} = 0$ and $\frac{\partial^2 z_{t,l}^c}{\partial \hat{z}_{t,i}^c \partial \hat{z}_{t,j}^c} = 0$ and $\frac{\partial z_{t,l}^{s_1}}{\partial \hat{z}_{t,i}^c} \cdot \frac{\partial z_{t,l}^{s_1}}{\partial \hat{z}_{t,j}^c} = 0$ and $\frac{\partial^2 z_{t,l}^{s_1}}{\partial \hat{z}_{t,i}^c \partial \hat{z}_{t,j}^c} = 0$. According to Equation (36),we have:

$$\mathbf{J}_{h_1} = \left[ \begin{array}{c|c} \mathbf{A} := \frac{\partial z_t^c}{\partial \hat{z}_t^c} & \mathbf{B} := \frac{\partial z_t^c}{\partial \hat{z}_t^{s_1}} = 0 \\ \hline \mathbf{C} := \frac{\partial z_t^{s_1}}{\partial \hat{z}_t^c} = 0 & \mathbf{D} := \frac{\partial z_t^{s_1}}{\partial \hat{z}_t^{s_1}} \end{array} \right]. \tag{45}$$

Since $h_1$ is invertible and for $i, j \in \{1, \cdots, n_c\}$, $\frac{\partial z_{t,l}^c}{\partial \hat{z}_{t,i}^c} \cdot \frac{\partial z_{t,l}^c}{\partial \hat{z}_{t,j}^c} = 0$ and $\frac{\partial z_{t,l}^{s_1}}{\partial \hat{z}_{t,i}^c} \cdot \frac{\partial z_{t,l}^{s_1}}{\partial \hat{z}_{t,j}^c} = 0$ implies that for each $k = 1, \cdots, n_c$, there is exactly one non-zero component in each column of matrices $\mathbf{A}$ and $\mathbf{C}$. Since we have proved that $\hat{z}_t^c = h_c(z_t^c)$ and $\mathbf{C} = 0$, there is exactly one non-zero component in each column of matrices $\mathbf{A}$. Therefore, $z_t^c$ is component-wise identifiable.

Based on Equation(40), we further let $i, j, k \in \{n_c + 1, \cdots, n\}$, and its three-order derivation w.r.t. $\hat{z}_{t,i}^{s_1}, \hat{z}_{t,j}^{s_1}, z_{t-1,l}^{s_1}$ can be written as

$$
\begin{aligned}
&\sum_{l=1}^{n_c} \frac{\partial^3 \log p(z_{t,l}^c | z_{t-1}^c)}{\partial^2 z_{t,l}^c \partial z_{t-1,k}^{s_1}} \cdot \frac{\partial z_{t,l}^c}{\partial \hat{z}_{t,j}^{s_1}} \cdot \frac{\partial z_{t,l}^c}{\partial \hat{z}_{t,i}^{s_1}} \\
&+ \sum_{l=1}^{n_c} \frac{\partial^2 \log p(z_{t,l}^c | z_{t-1}^1)}{\partial z_{t,l}^c \partial z_{t-1,k}^{s_1}} \cdot \frac{\partial^2 z_{t,l}^c}{\partial \hat{z}_{t,i}^{s_1} \partial \hat{z}_{t,j}^{s_1}} \\
&+ \sum_{l=n_c+1}^{n} \frac{\partial^3 \log p(z_{t,l}^{s_1} | z_{t-1}^1, z_t^c)}{\partial^2 z_{t,l}^{s_1} \partial z_{t-1,k}^{s_1}} \cdot \frac{\partial z_{t,l}^{s_1}}{\partial \hat{z}_{t,j}^{s_1}} \cdot \frac{\partial z_{t,l}^{s_1}}{\partial \hat{z}_{t,i}^{s_1}} \\
&+ \sum_{l=n_c+1}^{n} \frac{\partial^2 \log p(z_{t,l}^{s_1} | z_{t-1}^1, z_t^c)}{\partial z_{t,l}^{s_1} \partial z_{t-1,k}^{s_1}} \cdot \frac{\partial^2 z_{t,l}^{s_1}}{\partial \hat{z}_{t,i}^{s_1} \partial \hat{z}_{t,j}^{s_1}} = 0.
\end{aligned}
\tag{46}
$$

By using the linear independence assumption, the linear system denoted by Equation (44) has the only solution $\frac{\partial z_{t,l}^c}{\partial \hat{z}_{t,i}^{s_1}} \cdot \frac{\partial z_{t,l}^c}{\partial \hat{z}_{t,j}^{s_1}} = 0$ and

$\frac{\partial^2 z_{t,l}^c}{\partial \hat{z}_{t,i}^{s_1} \partial \hat{z}_{t,j}^{s_1}} = 0$ and $\frac{\partial z_{t,l}^{s_1}}{\partial \hat{z}_{t,i}^{s_1}} \cdot \frac{\partial z_{t,l}^{s_1}}{\partial \hat{z}_{t,j}^{s_1}} = 0$ and $\frac{\partial^2 z_{t,l}^{s_1}}{\partial \hat{z}_{t,i}^{s_1} \partial \hat{z}_{t,j}^{s_1}} = 0$, meaning that there is exactly one non-zero component in each row of $\mathbf{B}$ and $\mathbf{D}$. Since $\mathbf{B} = 0$ ,then $z_t^{s_1}$ is component-wise identifiable. Similarly, we can prove that $z_t^{s_m}$ is component-wise identifiable.

$\square$

## B Evidence Lower Bound

In this subsection, we show the evidence lower bound. We first factorize the conditional distribution according to the Bayes theorem.

$$
\ln p(x_{1:T}^{s_1}, x_{1:T}^{s_2}) = \ln \frac{p(x_{1:T}^{s_1}, x_{1:T}^{s_2}, z_{1:T}^{s_1}, z_{1:T}^{s_2}, z_{1:T}^c)}{p(z_{1:T}^{s_1}, z_{1:T}^{s_2}, z_{1:T}^c | x_{1:T}^{s_1}, x_{1:T}^{s_2})}
$$

$$
= \ln \frac{p(x_{1:T}^{s_1} | z_{1:T}^{s_1}, z_{1:T}^c) p(x_{1:T}^{s_2} | z_{1:T}^{s_2}, z_{1:T}^c) p(z_{1:T}^{s_1} | z_{1:T}^c) p(z_{1:T}^{s_2} | z_{1:T}^c) p(z_{1:T}^c)}{p(z_{1:T}^{s_1} | x_{1:T}^{s_1}, z_{1:T}^c) p(z_{1:T}^{s_2} | x_{1:T}^{s_2}, z_{1:T}^c)) p(z_{1:T}^c | x_{1:T}^{s_1}, x_{1:T}^{s_2})}
$$

$$
= \mathbb{E}_{q(z_{1:T}^{s_1} | x_{1:T}^{s_1}, z_{1:T}^c)} \mathbb{E}_{q(z_{1:T}^{s_2} | x_{1:T}^{s_2}, z_{1:T}^c)} \mathbb{E}_{q(z_{1:T}^c | x_{1:T}^{s_1}, x_{1:T}^{s_2})}
$$

$$
\cdot \ln \frac{p(x_{1:T}^{s_1} | z_{1:T}^{s_1}, z_{1:T}^c) p(x_{1:T}^{s_2} | z_{1:T}^{s_2}, z_{1:T}^c) p(z_{1:T}^{s_1} | z_{1:T}^c) p(z_{1:T}^{s_2} | z_{1:T}^c)}{q(z_{1:T}^{s_1} | x_{1:T}^{s_1}, z_{1:T}^c) q(z_{1:T}^{s_2} | x_{1:T}^{s_2}, z_{1:T}^c) q(z_{1:T}^c | x_{1:T}^{s_1}, x_{1:T}^{s_2})}
$$

$$
+ D_{KL}(q(z_{1:T}^{s_1} | x_{1:T}^{s_1}, z_{1:T}^c) || p(z_{1:T}^{s_1} | x_{1:T}^{s_1}, z_{1:T}^c)) + D_{KL}(q(z_{1:T}^{s_2} | x_{1:T}^{s_2}, z_{1:T}^c) || p(z_{1:T}^{s_2} | x_{1:T}^{s_2}, z_{1:T}^c))
$$

$$
+ D_{KL}(q(z_{1:T}^c | x_{1:T}^{s_1}, x_{1:T}^{s_2}) || p(z_{1:T}^c | x_{1:T}^{s_1}, x_{1:T}^{s_2}))
$$

$$
\geq \mathbb{E}_{q(z_{1:T}^{s_1} | x_{1:T}^{s_1}, z_{1:T}^c)} \mathbb{E}_{q(z_{1:T}^{s_2} | x_{1:T}^{s_2}, z_{1:T}^c)} \mathbb{E}_{q(z_{1:T}^c | x_{1:T}^{s_1}, x_{1:T}^{s_2})}
$$

$$
\cdot \ln \frac{p(x_{1:T}^{s_1} | z_{1:T}^{s_1}, z_{1:T}^c) p(x_{1:T}^{s_2} | z_{1:T}^{s_2}, z_{1:T}^c) p(z_{1:T}^c) p(z_{1:T}^{s_1} | z_{1:T}^c) p(z_{1:T}^{s_2} | z_{1:T}^c)}{q(z_{1:T}^{s_1} | x_{1:T}^{s_1}, z_{1:T}^c) q(z_{1:T}^{s_2} | x_{1:T}^{s_2}, z_{1:T}^c) q(z_{1:T}^c | x_{1:T}^{s_1}, x_{1:T}^{s_2})}
$$

$$
= \underbrace{\mathbb{E}_{q(z_{1:T}^{s_1} | x_{1:T}^{s_1}, z_{1:T}^c)} \mathbb{E}_{q(z_{1:T}^{s_2} | x_{1:T}^{s_2}, z_{1:T}^c)} \mathbb{E}_{q(z_{1:T}^c | x_{1:T}^{s_1}, x_{1:T}^{s_2})} \ln p(x_{1:T}^{s_1} | z_{1:T}^{s_1}, z_{1:T}^c) p(x_{1:T}^{s_2} | z_{1:T}^{s_2}, z_{1:T}^c)}_{\mathcal{L}_{rec}}
$$

$$
\underbrace{- D_{KL}(q(z_{1:T}^{s_1} | x_{1:T}^{s_1}, z_{1:T}^c) || p(z_{1:T}^{s_1} | z_{1:T}^c))}_{\mathcal{L}_{s_1}} \underbrace{- D_{KL}(q(z_{1:T}^{s_1} | x_{1:T}^{s_2}, z_{1:T}^c) || p(z_{1:T}^{s_2} | z_{1:T}^c))}_{\mathcal{L}_{s_2}}
$$

$$
\underbrace{- D_{KL}(q(z_{1:T}^c | x_{1:T}^{s_1}, x_{1:T}^{s_2}) || p(z_{1:T}^c))}_{\mathcal{L}_c}
$$

$$
\tag{47}
$$

## C Prior Estimation

**Shared Prior Estimation:** We first consider the prior of $\ln p(z_{1:T}^c)$. We consider the time lag as $L = 1$,we devise a transformation $\sigma^c := \{\hat{z}_{t-1}^c, \hat{z}_t^c\} \rightarrow \{\hat{z}_{t-1}^c, \hat{\epsilon}_t^c\}$. Then we write this latent process as a transformation map $\sigma$ (note that we overload the notation $\sigma$ for transition functions and for the transformation map):

$$
\begin{bmatrix} \hat{z}_{t-1}^c \\ \hat{z}_t^c \end{bmatrix} = \sigma\left( \begin{bmatrix} \hat{z}_{t-1}^c \\ \hat{\epsilon}_t^c \end{bmatrix} \right).
$$

By applying the change of variables formula to the map $\mathbf{f}$, we can evaluate the joint distribution of the latent variables $p(\hat{z}_{t-1}^c \hat{z}_t^c)$ as

$$
p(\hat{z}_{t-1}^c, \hat{z}_t^c) = \frac{p(\hat{z}_{t-1}^c \hat{\epsilon}_t^c)}{|\det \mathbf{J}_\sigma|},
\tag{48}
$$

where $\sigma_\sigma$ is the Jacobian matrix of the map $\mathbf{f}$, which is naturally a low-triangular matrix:

$$
\mathbf{J}_\sigma = \begin{bmatrix} 1 & 0 \\ \frac{\partial \hat{z}_t^c}{\partial \hat{z}_{t-1}^c} & \frac{\hat{z}_t^c}{\hat{\epsilon}_t^c} \end{bmatrix}.
$$

Let $\{r_i^c\}_{i=1,2,3,\cdots}$ be a set of learned inverse transition functions that take the estimated latent causal variables, and output the noise terms, i.e., $\hat{\epsilon}_{t,i} = r_i^c(\hat{z}_{t,i}^c, \hat{z}_{t-1}^c)$. Then we design a transformation $\mathbf{A} \rightarrow \mathbf{B}$ with low-triangular Jacobian as follows:

$$
\underbrace{[\hat{z}_{t-1}^c, \hat{z}_t^c]^\top}_{\mathbf{A}} \text{ mapped to } \underbrace{[\hat{z}_{t-1}^c, \hat{\epsilon}_t^c]^\top}_{\mathbf{B}}, \text{ with } \mathbf{J}_{\mathbf{A} \rightarrow \mathbf{B}} = \begin{bmatrix} \mathbb{I} & 0 \\ * & \text{diag}\left( \frac{\partial r_i^c}{\partial \hat{z}_{t-1,i}^c} \right) \end{bmatrix}.
\tag{49}
$$

Similar to Equation (49), we can obtain the joint distribution of the estimated dynamics subspace as:

$$
\log p(\mathbf{A}) = \log p(\mathbf{B}) + \log(|\det(\mathbf{J}_{\mathbf{A} \rightarrow \mathbf{B}})|).
\tag{50}
$$

Finally, we have:

$$
\log p(\hat{z}_t^c | z_{t-1}^c) = \log p(\hat{\epsilon}_t^c) + \sum_{i=n_d+1}^{n} \log | \frac{\partial r_i^c}{\partial \hat{z}_{t-1,i}^c} |.
\tag{51}
$$

As a result, the prior distribution shared latent variables can be estimated as follows:

$$
p(\hat{z}_{1:T}^c) = p(\hat{z}_1^c) \prod_{\tau=2}^{T} \left( \sum_{i=n_d+1}^{n} \log p(\hat{\epsilon}_{\tau,i}^c) + \sum_{i=n_d+1}^{n} \log | \frac{\partial r_i^c}{\partial \hat{z}_{\tau-1,i}^c} | \right),
\tag{52}
$$

where we assume $p(\hat{\epsilon}_{\tau,i}^c)$ follows a standard Gaussian distribution.

As for the modality-specific prior estimation, we can obtain a similar derivation, by considering the modality-shared prior as a condition.

## D Implementation Details

We summarize our network architecture below and describe it in detail in Table 7. We also provide the training details in Table 8 and 9. Moreover, we provide a statistical summary of the evaluated dataset in Table10.

## E Experiment Details

### E.1 Dataset Descriptions

In this paper, we consider the WIFI [104], and KETI [24] datasets. Moreover, we further consider the human motion prediction datasets like Motion [82], HumanEva-I [87], H36M [34], UCIHAR [1], PAMAP2

**Table 7: Architecture details. BS: batch size, T: length of time series, LeakyReLU: Leaky Rectified Linear Unit, $|x_t|$: the dimension of $x_t$.**

| Configuration | Description | Output |
|---|---|---|
| 1. $\psi_c$ | Modality-shared Encoder | |
| Input:$x_{1:T}$ | Observed time series | BS $\times t \times |x_T|$ |
| Augmentations | Time-Domain Transpose | BS $\times 2 t \times |x_T|$ |
| CNN Block | 150 neurons | BS $\times t \times 150$ |
| CNN Block | 150 neurons | BS $\times t \times 150$ |
| Permute | Matrix Transpose | BS $\times 150 \times t$ |
| GRU | 300 neurons | BS $\times 300$ |
| Split | Transpose | BS $\times t \times n_c$ |
| 2. $\psi_s$ | Modality-private Encoder | |
| Input:$x_{1:T}$ | Observed time series | BS $\times t \times |x_T|$ |
| Augmentations | Time-Domain Transpose | BS $\times 2 \times t \times |x_T|$ |
| CNN Block | 150 neurons | BS $\times t \times 150$ |
| CNN Block | 150 neurons | BS $\times t \times 150$ |
| Permute | Matrix Transpose | BS $\times 150 \times t$ |
| GRU | 300 neurons | BS $\times 300$ |
| Split | Transpose | BS $\times t \times n_c$ |
| Dense | $n_s$ neurons | BS $\times t \times n_s$ |
| 3.$F_x$ | Reconstruction Decoder | |
| Input:$z_{1:T}^c, z_{1:T}^s$ | Modality-share and Modality-privte Latent Variable | BS$\times t \times n_c$, BS$\times t \times n_s$ |
| Concat | concatenation | BS $\times t \times (n_c + n_s)$ |
| Dense | x dimension neurons | BS $\times t \times |x_T|$ |
| 4.$F_y$ | Downstream task Predictor | |
| Input:$z_{1:T}^c, z_{1:T}^s$ | Modality-share and Modality-private Latent Variable | BS $\times t \times n_c$ ,BS$\times t \times n_s$ |
| Concat | concatenation | BS $\times t \times (n_s + n_c)$ |
| Dense | x neurons,GELU | BS $\times t \times x$ |
| Dense | n neurons | BS $\times t \times n$ |
| 5.$r_c$ | Modality-share Prior Networks | |
| Input:$z_{1:T}^c$ | Latent Variables | BS $\times (n_c + 1)$ |
| Dense | 128 neurons,LeakyReLU | $(n_c + 1) \times 128$ |
| Dense | 128 neurons,LeakyReLU | $128 \times 128$ |
| Dense | 128 neurons,LeakyReLU | $128 \times 128$ |
| Dense | 1 neuron | BS $\times 1$ |
| JacobianCompute | Compute log ( abs(det (J))) | BS |
| 6.$r_s$ | Modality-private Prior Networks | |
| Input:$z_{1:T}^s$ and $z_{1:T}^c$ | Latent Variables | BS $\times (n_c + n_s + 1)$ |
| Dense | 128 neurons,LeakyReLU | $(n_c + n_s + 1) \times 128$ |
| Dense | 128 neurons,LeakyReLU | $128 \times 128$ |
| Dense | 128 neurons,LeakyReLU | $128 \times 128$ |
| Dense | 1 neuron | BS $\times 1$ |
| JacobianCompute | Compute log (abs( det (J))) | BS |

**Table 8: Supervised Training Congfigurations(We use LR for Learning Rate).**

| Dataset | Motion | DINAMO | WIFI | KETI | HumanEVA | H36M | MIT-BIH | UCIHAR | HAC | EPIC-Kitchens |
|---|---|---|---|---|---|---|---|---|---|---|
| Temperature | 0.5 | 0.5 | 0.5 | 0.5 | 0.5 | 0.5 | 0.5 | 0.5 | 0.5 | 0.5 |
| Batch Size | 32 | 64 | 32 | 64 | 64 | 64 | 64 | 64 | 64 | 64 |
| Window Length | 256 | 256 | 256 | 256 | 75 | 125 | 64 | 128 | 256 | 256 |
| Supervised Optimizer | AdawW | AdawW | AdawW | AdawW | AdawW | AdawW | AdawW | AdawW | AdawW | AdawW |
| Supervised Max LR | 1e-4 | 1e-4 | 1e-4 | 1e-4 | 1e-4 | 1e-4 | 1e-4 | 1e-4 | 1e-4 | 1e-4 |
| Supervised Min LR | 1e-6 | 1e-6 | 1e-6 | 1e-6 | 1e-6 | 1e-6 | 1e-6 | 1e-6 | 1e-6 | 1e-6 |
| Supervised Scheduler | cosine | cosine | cosine | cosine | cosine | cosine | cosine | cosine | cosine | cosine |

[81], and RealWorld-HAR [89]which consider different positions of the human body as different modalities. Moreover, we also consider two healthcare datasets such as MIT-BIH [72] and D1NAMO [17], which are related to arrhythmia and noninvasive type 1 diabetes. Moreover, we have further considered audio and video datasets, such as HAC[15] and EPIC-Kitchens[9], which include three modalities: video, audio, and pre-computed optical flow.

**Motion** [82] dataset is a subset of the OPPORTUNITY Activity Recognition Dataset [82]. Following the experimental setting of a recent device-based HAR study [36], we consider 5 sensors worn at 5 different locations on the human body: left lower arm, left upper arm, right lower arm, right upper arm and the back. Each device contains an accelerometer, a gyroscope, and a magnetometer, and all three sensors generate three-axis readings. We focus on a 4-class

**Table 9: Self-Supervised Training Congfigurations(We use LR for Learning Rate).**

| Dataset | UCIHAR | RealWorld-HAR | PAMAP2 | MIT-BIH | D1NAMO | KETI |
|---|---|---|---|---|---|---|
| Temperature | 0.5 | 0.5 | 0.5 | 0.5 | 0.5 | 0.5 |
| Batch Size | 64 | 64 | 64 | 64 | 64 | 64 |
| Window Length | 128 | 150 | 512 | 64 | 256 | 256 |
| Supervised Optimizer | AdawW | – | – | AdawW | AdawW | AdawW |
| Supervised Max LR | 1e-4 | – | – | 1e-4 | 1e-4 | 1e-4 |
| Supervised Min LR | 1e-6 | – | – | 1e-6 | 1e-6 | 1e-6 |
| Supervised Scheduler | cosine | – | – | cosine | cosine | cosine |
| Pretrain Optimizer | AdawW | AdawW | AdawW | AdawW | AdawW | AdawW |
| Pretrain Max LR | 1e-3 | 1e-3 | 1e-3 | 1e-4 | 1e-4 | 1e-4 |
| Pretrain Min LR | 1e-7 | 1e-7 | 1e-7 | 1e-7 | 1e-7 | 1e-7 |
| Pretrain Scheduler | cosine | cosine | cosine | cosine | cosine | cosine |
| Pretrain Weight Decay | 0.5 | 0.5 | 0.5 | 0.5 | 0.5 | 0.5 |
| Finetune Optimizer | AdawW | – | – | – | – | – |
| Finetune Start LR | 1e-3 | – | – | – | – | – |
| Finetune Scheduler | cosine | – | – | – | – | – |
| Finetune LR Decay | 0.2 | – | – | – | – | – |
| Finetune LR Period | 50 | – | – | – | – | – |
| Finetune Epochs | 200 | – | – | – | – | – |

**Table 10: Statistical Summaries of Evaluated Datasets.**

| Dataset | Modalities | Windows | Classes |
|---|---|---|---|
| Motion | 5x(Acc,Gyro,Mag) (back, left L-arm, right U-arm, left/right shoe) | 256 | 4 |
| D1NAMO | ECG(lead II, lead V1) | 256 | 2 |
| WIFI | Wireless x3 | 256 | 7 |
| KETI | 4 sensors (monitoring CO2, temperature, humidity and light intensity) | 256 | 2 |
| HumanEVA | Skeleton x15 | 75 | 5 |
| H36M | Skeleton x17 | 125 | 15 |
| UCIHAR | Body ACC,Total ACC, Total Gyro | 128 | 6 |
| MIT-BIH | Heart Rate, Breathing Rate, Avg Acceleration, Peak Acceleration | 64 | 5 |
| Relative-World | acc, gyro, mag | 150 | 8 |
| PAMAP2 | acc, gyro | 512 | 18 |
| EPIC-Kitchens | video, audio, flow | 256 | 8 |
| HAC | video, audio, flow | 256 | 7 |

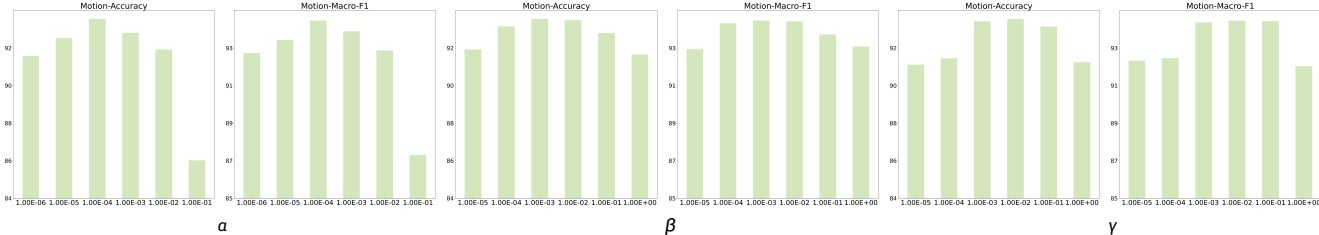

**Figure 6: Experiments results of different values of $\alpha$,$\beta$, $\gamma$ on Motion dataset.**

prediction consisting of high-level locomotion activities (sit, stand, walk and lie).

**D1NAMO** [17] is acquired on 20 healthy subjects and 9 patients with type-1 diabetes. The acquisition has been made in real-life conditions with the Zephyr BioHarness 3 wearable device. The dataset consists of ECG, breathing, and accelerometer signals, as well as glucose measurements and annotated food pictures.

**WIFI** [104] dataset contains the amplitude and phase of wireless signals sent by three antennas. Each antenna transmits at 30 sub-carriers, and the receiver base sampling frequency is 1000 Hz. The

dataset contains 7 classes of activity, including lying down, falling, picking up, running, sitting down, standing up and walking. We also use a sliding window of 256 timestamps to get the segmented examples.

**KETI** [24]dataset was collected from 51 rooms in a large university office building. Each room is instrumented with 4 sensors monitoring CO2, temperature, humidity and light intensity, with occupancy monitored by an additional PIR sensor in the room. Readings are recorded every 10 seconds, and the dataset contains

one week worth of data. In this experiment, we target at human occupation prediction using the readings of these sensors.

**HumanEVA-I** [87] comprises 3 subjects each performing 5 actions. We apply the original frame rate (60 Hz) and a 15-joint skeleton removing the root joint to build human motions.

**H36M** [34] consists of 7 subjects (S1, S5, S6, S7, S8 ,S9 and S11) performing 15 different motions. We apply the original frame rate (50 Hz) and a 17-joint skeleton removing the root joint to build human motions.

**UCIHAR** [1] dataset contains recordings from 30 volunteers who carried out 6 classes of activities, including walking, walking upstairs, walking downstairs, sitting, standing, and lying. Activities are recorded by a smartphone device mounted on the volunteer's waist.

**MIT-BIH** [72] contains 48 records obtained from 47 subjects. Each subject is represented by one ECG recording using two leads: lead II (MLII) and lead V1. The sampling frequency of the signal is 360 Hz. The upper signal is lead II (MLII) and the lower signal is lead V1, obtained by placing the electrodes on the chest. In the upper signal, the normal QRS complexes are usually prominent.

**RealWorld-HAR** [89] is a public dataset using an accelerometer, gyroscope, magnetometer, and light signals from the forearm, thigh, head, upper arm, waist, chest, and shin to recognize eight common human activities performed by 15 subjects, including climbing stairs down and up, jumping, lying, standing, sitting, running/jogging, and walking.In our experiments, we only used the data collected from the "waist" sensor, including the accelerometer (ACC) and gyroscope. The sampling rate for all selected sensors was set at 100Hz.

**PAMAP2** [81] contains data on 18 different classes of physical activities performed by 9 subjects wearing 3 inertial measurement units and a heart rate monitor. In this set of experiments, we only used 3 accelerometer sensor data and 18 activities. Only data collected from the "wrist" is used in our experiment

**HAC** [15] contains two multi-modality datasets named Human and Cartoon, which contain three modalities: video, audio, and pre-computed optical flow. The human and cartoon datasets contain seven actions ('sleeping', 'watching TV', 'eating', 'drinking', 'swimming', 'running', and 'opening door'), which are a subset of the HAC dataset.

**EPIC-Kitchens** [9] contain multi-modality datasets D2, which contain three modalities: video, audio, and pre-computed optical flow. The D2 dataset contains eight actions ('put ',' take ',' open ',' close ',' wash ',' cut ',' mix ', and' pour ') recorded in three different kitchens, and is a subset of the EPIC Kitchens dataset.

## E.2 More Experiment Results

*E.2.1 Sensitivity Analysis.* Figure 6 provides the results of sensitivity analysis.

## F Limitation

Although our method can learn disentangled representation for multi-modal time series data with identifiability guarantees, it requires the assumption that the mixing function is invertible. However, this assumption might be hard to meet in real-world scenarios. Therefore, how to leverage the temporal context information to address this challenge will be an interesting direction.

