# OpenReview forum: "Learning Disentangled Representation for Multi-Modal Time-Series Sensing Signals"
_ACM.org/TheWebConf/2025/Conference — WWW 2025 Poster_

### Official Review · Reviewer_jEgo · 2024-11-06

**Novelty:** 4
**Technical Quality:** 6

**Review:**

The paper presents a novel model, Multi-modal Temporal Disentanglement (MATE), for effectively disentangling modality-shared and modality-specific latent variables in multi-modal time-series data. Based on the data generation process for multi-modal time series data with dependent latent subspace, it devises a general disentangled representation learning framework with identifiability guarantees. Compared with the existing methods, the proposed MATE model can learn the disentangled time series representations even in the dependent latent subspace.

Pros：

1 The MATE model advances beyond traditional orthogonal approaches by modeling modality-shared and modality-specific latent variables as dependent. This more realistic representation increases its applicability to real-world datasets with complex inter-modal relationships.

2 The paper provides rigorous theoretical results to establish subspace and component-wise identifiability of the disentangled representations, a significant contribution that ensures the reliability of the learned latent variables.

Cons:

1 Excessive Use of Equations in the Main Text: While the paper is thorough in its formalism, the large number of equations can make it challenging to follow all technical details. Although the logical flow between equations is generally clear, some technical nuances may be difficult to grasp. Consider moving some equations to an appendix to enhance readability.

2 Lack of Explicit Connection Between Sections 4 and 5: The relationship between Section 4 (the framework) and Section 5 (theoretical analysis) is not clearly articulated. Currently, it appears that the theoretical analysis in Section 5 does not directly impact the design of the framework in Section 4 or the subsequent experimental design, though there likely is a strong connection. It would benefit readers if the authors emphasized this link.

3 Formatting and Clarity Issues: There are minor inconsistencies in formatting and expression. For example, some equations include punctuation at the end while others do not. Additionally, the Jacobian symbol on line 403 requires clarification before use, and Tables 1 and 2 lack self-explanatory details as the titles do not clarify the meaning of values in parentheses.

**Questions:**

see weaknesses.

**Reviewer Confidence:**

2: The reviewer is willing to defend the evaluation, but it is likely that the reviewer did not understand parts of the paper

**Scope:**

3: The work is somewhat relevant to the Web and to the track, and is of narrow interest to a sub-community

---

### Official Review · Reviewer_BfoY · 2024-11-25

**Novelty:** 2
**Technical Quality:** 3

**Review:**

In this paper, a multimodal temporal disentanglement (MATE) model is proposed for learning the representation of disentanglement from multimodal time series data. Existing multimodal time series representation learning methods usually assume that the potential space is orthogonal, but in real-world scenarios, there may be dependencies between modal specific and modal shared latent variables.

Pros

1.A Multi-modAl TEmporal Disentanglement (MATE) model is proposed, which can learn the decoupling representation from multimodal time series data.

2.Numerous experiments on 12 datasets.

Cons

1.This paper lacks novelty. There has been a lot of research on variational disentanglement on potential distributions. This work looks like applying them to multimodal time series.

2.The MATE model requires the assumption that the mixing function is reversible, but this assumption may be difficult to satisfy in real-world scenarios.

**Questions:**

See Cons.

**Reviewer Confidence:**

3: The reviewer is confident but not certain that the evaluation is correct

**Scope:**

3: The work is somewhat relevant to the Web and to the track, and is of narrow interest to a sub-community

---

### Official Review · Reviewer_mvyF · 2024-11-27

**Novelty:** 5
**Technical Quality:** 5

**Review:**

The paper proposes MATE that is built on a temporally variational inference architecture with the modality-shared and modality-specific prior networks for the dis-entanglement of latent variable. The model is novel and achieves comparable performance.

Strengths:
1. The proposed problem, the orthogonality of modality-shared and modality-specific latent space may be too difficult to satisfy in real-world scenarios, has research significance.
2. The proposed model is novel, however, it seems unrelated to the problem mentioned above and shares the key idea with prior disentangled methods, that is, seperating the shared and specific components.

Limitations:

1. The section of Introduction needs to be reorganized, I can not find a clear and key motivation. Authors claim that the orthogonality of modality-shared and modality-specific latent space may be too difficult to satisfy in real-world scenarios, which is a interesting and realistic problem. However, I cann't find any further explore for this issuse that should be a important motivation. Moreover, authors state that other methods use the orthogonality to achieve disentanglement, but write "existing methods that apply orthogonal constraints on the estimated latent variables despite the dependent true latent sources, lead to the entanglement", which confues me.
2. There are many and complex symbols, it is recommended to provide a summary table.
3. Authors seems to have not provided the source code to enhance their study.

**Questions:**

1.The manuscript title seems to be inconsistent with the submission title. The authors appear to have used an improper writing template, and it seems to violate the double-blind agreement.
2. The section of Introduction needs to be reorganized, I can not find a clear and key motivation. Authors claim that the orthogonality of modality-shared and modality-specific latent space may be too difficult to satisfy in real-world scenarios, which is a interesting and realistic problem. However, I cann't find any further explore for this issuse that should be a important motivation. Moreover, authors state that other methods use the orthogonality to achieve disentanglement, but write "existing methods that apply orthogonal constraints on the estimated latent variables despite the dependent true latent sources, lead to the entanglement", which confues me.
3. There are many and complex symbols, it is recommended to provide a summary table.
4.The illustrations in the manuscript may not be standardized and lack reasonable explanations for the symbols. For instance, the symbols in Equation (13) do not match those in Figure 3. It seems that the authors need to optimize the images inserted into the manuscript.
4. Authors seems to have not provided the source code to enhance their study.

**Reviewer Confidence:**

4: The reviewer is certain that the evaluation is correct and very familiar with the relevant literature

**Scope:**

4: The work is relevant to the Web and to the track, and is of broad interest to the community

---

### Official Review · Reviewer_VqEp · 2024-11-29

**Novelty:** 4
**Technical Quality:** 4

**Review:**

**Summary:**

This paper introduces the MATE model for multi-modal time series data, addressing the dependency between modality-shared and modality-specific latent variables often observed in real-world scenarios. Built on a temporally variational inference framework, MATE ensures disentanglement through identifiable latent representations. Experiments on 12 datasets demonstrate its effectiveness across various downstream tasks.

**Strengths:**

1. The paper addresses time series representation, a critical and widely studied research problem.
2. The experimental evaluation is conducted on a diverse and comprehensive set of real datasets.
3. The results demonstrate strong performance, particularly on classification tasks.

**Weaknesses:**

See questions.

**Questions:**

1. The paper lacks clarity in defining what constitutes multi-modal data, such as which variables belong to a specific modality and how this concept differs from or relates to multivariate time series data. Additionally, Section 3.1 does not specify the dimensionality of each modal variable, which warrants a more detailed explanation.
2. The scalability of the proposed model to scenarios with more than two modalities is unclear. How would the model handle an increased number of modalities in terms of spatial and temporal complexity?
3. Regarding the assumptions in Section 5.2, particularly Assumption 3, is there empirical evidence from the datasets to validate this claim?
4. The baselines compared in the experiments lack representation from more recent time series representation learning methods or foundational models, such as those proposed in [1][2][3]. Including these would strengthen the comparative analysis.
5. The experiments focus exclusively on classification tasks. How does the proposed method perform on other downstream tasks in the time series domain, such as forecasting, anomaly detection, imputation, or clustering? Including such analyses would provide a more comprehensive evaluation of the method's versatility.

[1] Dong J, Wu H, Zhang H, et al. Simmtm: A simple pre-training framework for masked time-series modeling[J]. Advances in Neural Information Processing Systems, 2024, 36.

[2] Zhou T, Niu P, Sun L, et al. One fits all: Power general time series analysis by pretrained lm[J]. Advances in neural information processing systems, 2023, 36: 43322-43355.

[3] Lee S, Park T, Lee K. Learning to Embed Time Series Patches Independently[C]//The Twelfth International Conference on Learning Representations.

**Reviewer Confidence:**

3: The reviewer is confident but not certain that the evaluation is correct

**Scope:**

3: The work is somewhat relevant to the Web and to the track, and is of narrow interest to a sub-community

---

### Official Review · Reviewer_WW2u · 2024-12-02

**Novelty:** 5
**Technical Quality:** 5

**Review:**

This manuscript targets the time series prediction task. It claims that current methods in the field disentangle the modality-specific and modality-shared latent variables, i.e., the assumption of an orthogonal latent space. Though they have made some progress, the assumption may not coincide with real-world scenarios. Thus, the authors propose that modality-specific and modality-shared latent variables are dependent. Further, they progressively establish the identifiability of subspaces and component-wise modality-specific latent variables. Specifically, a temporally variational inference architecture is utilized to disentangle the latent variables. To validate the effectiveness of the proposed method, both theoretical and experimental results are carried out.

Pros:

- The motivation of the manuscript that modality-shared and modality-specific latent variables may not orthogonal is reasonable. The proposed method, which explicitly models the dependence between these two types of variables, corresponds to the motivation.
- The theoretical results in section 5 states that the disentangled representation can be learned by the proposed method, which seems sound. The experiments on 3 types of tasks show that the proposed method achieves state-of-the-art on all datasets.
- The paper is generally well-written and easy to follow for the most part. The visualization of the framework overview and the detailed information transfer are clear. The usage of terms is accurate and consistent.

Cons:

- In the manuscript, the proposed method is expected to work in scenarios more than 2 modalities. But in my understanding, the experiments are all conducted on datasets which only involves two modalities. Intuitively, the number of modalities may have significant influence on the method design and performance. Thus, it would be better to discuss the robustness of the proposed method against the number of modalities.
- In section 6.3, the manuscript visualizes the domain-shared latent variables. Though the proposed method forms clear clusters, some baselines such as COCOA and CROSSL also exhibit clear boundaries between clusters. Thus, the manuscript would benefit from a more through analysis and more implementation details in this subsection.

**Questions:**

Please refer to the Cons in the above part.

**Reviewer Confidence:**

2: The reviewer is willing to defend the evaluation, but it is likely that the reviewer did not understand parts of the paper

**Scope:**

3: The work is somewhat relevant to the Web and to the track, and is of narrow interest to a sub-community